# Assessing the performance of global hydrological models for capturing peak river flows in the Amazon Basin

Jamie Towner[1], Hannah L. Cloke[1,2,4,5], Ervin Zsoter[3,1], Zachary Flamig[6], Jannis M. Hoch[7,8], Juan Bazo[10,11], Erin Coughlan de Perez[9,10], Elisabeth M. Stephens[1]

[1]Department of Geography & Environmental Science, University of Reading, Reading, RG6 6AB, UK
[2]Department of Meteorology, University of Reading, Reading, RG6 6BB, UK
[3]European Centre for Medium-Range Weather Forecasts, Shinfield Park, Reading, RG6 9AX, UK
[4]Department of Earth Sciences, Uppsala University, Uppsala, 752 36, Sweden
[5]Centre of Natural Hazards and Disaster Science, CNDS, Uppsala, 752 36, Sweden
[6]University of Chicago Center for Data Intensive Science, Chicago, USA
[7]Department of Physical Geography, Utrecht University, P.O. Box 80115, 3508 TC Utrecht, the Netherlands
[8]Deltares, P.O. Box 177, 2600 MH Delft, the Netherlands
[9]International Research Institute for Climate and Society, Columbia University, Palisades, NY 10964, USA
[10]Red Cross Red Crescent Climate Centre, The Hague, 2521 CV, the Netherlands
[11] Universidad Tecnológica del Perú (UTP), Lima, Perú

*Correspondence to*: Jamie Towner (j.towner@pgr.reading.ac.uk)

**Abstract.** Extreme flooding impacts millions of people that live within the Amazon floodplain. Global Hydrological Models (GHMs) are frequently used to assess and inform the management of flood risk, but knowledge on the skill of available models is required to inform their use and development. This paper presents an intercomparison of eight different GHMs freely available from collaborators of the Global Flood Partnership (GFP) for simulating floods in the Amazon basin. To gain insight into the strengths and shortcomings of each model, we assess their ability to reproduce daily and annual peak river flows against gauged observations at 75 hydrological stations over a 19-year period (1997-2015). As well as highlighting regional variability in the accuracy of simulated streamflow these results indicate that a) the meteorological input is the dominant control on the accuracy of both daily and annual maximum river flows, and b) groundwater and routing calibration of Lisflood based on daily river flows has no impact on the ability to simulate flood peaks for the chosen river basin. These findings have important relevance for applications of large-scale hydrological models, including analysis of the impact of climate variability, assessment of the influence of long-term changes such as land-use and anthropogenic climate change, the assessment of flood likelihood, and for flood forecasting systems.

## 1 Introduction

Flooding is notably the most common and damaging natural hazard affecting millions of people worldwide every year, producing economic losses exceeding billions of dollars (Hirabayashi et al., 2012). Flood risk associated to a particular location can be highly variable depending on levels of exposure, resilience and preparedness (Alfieri et al., 2018) in addition to the increased uncertainty surrounding trends of hydrological extremes in a warming climate (Arnell and Gosling, 2016).

For the Amazon basin, flood risk is considered to have increased, with a greater frequency of extreme flood events (e.g. in 2009, 2012 and 2014; Marengo and Espinoza, 2016) coinciding with a hypothesised intensification of the hydrological cycle since the 1980's (Gloor et al., 2013). Floods in Amazonian communities are known to have large socioeconomic consequences impacting eco-systems, health, transport links and are particularly damaging to agricultural and fishery

practices (Schöngart and Junk 2007; Marengo et al., 2012; Marengo et al., 2013; Correa et al., 2017). Single flood events (e.g. 2012 in the Amazonian city of Iquitos, Peru) have impacted the lives of over 73,000 people (IFRC, 2013) with average annual damages estimated at 1.4 billion USD over a four-year period (2008-2011) in the Brazilian Rio Branco river basin alone (Mundial Grupo Banco, 2014).

## 1.1 Global hydrological models and applications

In its simplest form, a hydrological model can be considered a representation of a real-world hydrological system used to better understand various water and environmental processes, predict system behaviour and provide consistent impact assessment (Devia et al., 2015). They work by simulating the hydrological response to meteorological variations incorporating run-off generation and river routing processes (Sutanudjaja et al., 2018). As such, Global Hydrological Models (GHMs) have been used in a wide range of applications including short to extended-range flood forecasting (Alfieri et al.,

2013; Emerton et al., 2018), climate assessment (Hattermann et al., 2017), hazard and risk-mapping (Ward et al., 2016), drought prediction (van Huijevoort et al., 2014) and water resource assessment (e.g. water availability models; Meigh et al., 1999; Sood & Smakhtin, 2015).

Depending on the application and the needs of decision makers, different properties of the hydrograph simulated by hydrological models are important. For example, an accurate representation of peak river flows and their likelihood is key

for decision-makers who wish to understand the area at risk of flooding. In contrast, estimates of daily streamflow may be more beneficial for the assessment of water resources such as irrigation requirements.

## 1.2 GHM development

The availability of GHMs has grown in recent years thanks to increased efforts in addressing water related issues in developing countries (De Groeve et al., 2015; Ward et al., 2015; Trigg et al., 2016), the development of flood forecasting

systems (Aliferi et al., 2013; Werner et al., 2013; Emerton et al., 2018), improvements within precipitation datasets (Mittermaier et al., 2013; Novak et al., 2014; Forbes et al., 2015), the emergence of new global satellite and remote sensing datasets and advancements in numerical modelling techniques (Yamazaki et al., 2014a; Sampson et al., 2015; Andreadis et al., 2017; Balsamo et al., 2018). For an overview of available GHMs see Bierkens et al. (2015) who have provided the details of 22 large-scale hydrological models with those used for operational flood forecasting being summarised in Emerton

et al. (2016).

### 1.3 Land surface models vs hydrological models

GHMs have differing spatial and temporal resolutions, parameter estimation approaches, number of parameters, calibration methods, input-output variables and overall structures (Sood and Smakhtin, 2015). Their set-ups can generally be divided into two categories: Land Surface Models (LSMs) and hydrological models (Gudmundsson et al., 2012). The majority of

LSMs and hydrological models share the same conceptualisation of the water balance (Haddeland et al., 2011) but differ in their objective. LSMs evolve from coupled land/atmosphere models with the purpose of solving the surface energy balance equations to provide the necessary lower boundary conditions to the atmosphere (Wood et al., 2011). In contrast, hydrological models tend to focus less on the partitioning of radiation and more on hydrological resources and understanding the lateral movement and transport of water along the land surface.

In terms of differences in model performance, the Gudmundsson et al. (2012) intercomparison study of six LSMs and five GHMs (i.e. hydrological models) concluded that the main differences were due to the snow scheme implemented with snow water equivalent values and mean runoff fractions lower in LSMs. No significant differences between LSMs and hydrological models were found for runoff and evapotranspiration globally but rather the differences between the models themselves created large sources of uncertainty, highlighting the importance of analysing a range of different GHMs rather

than a group consisting of a specific model type. For the purposes of this study, we categorise both LSM and hydrological models as GHMs.

### 1.4 Motivation

For GHMs to be considered effective, end users need to know their accuracy and reliability (Ward et al., 2015). Thus, the evaluation of these models against observed data is an important procedure in efforts to reduce flood risk. Currently, no

intercomparison analysis of GHMs has been conducted specifically for the Amazon basin with previous studies focusing solely on the performance of individual models for the Amazon (e.g. Yamazaki et al., 2012; Paiva et al., 2013; Hoch et al., 2017a; Hoch et al., 2017b) or as part of a global study (e.g. Gudmundsson et al., 2012; Alfieri et al., 2013; Hirpa et al., 2018), which lack an in-depth focus on skill within the Amazon basin.

Finally, many of the GHMs (or their components) analysed in this study are used for specific applications. For instance, in

water resources management (PCRaster Global Water Balance; PCR-GLOBWB), flash flood forecasting (Ensemble Framework for Flash Flood Forecasting; EF5) and extended-range flood forecasting (Global Flood Awareness System; GloFAS). Investigating the performance of hydrological simulations therefore can provide valuable information to researchers and model developers with which to better understand some of the strengths and weaknesses which exists within the model set-ups and help to distinguish how different parts of the hydrological chain can cause particularly 'good' or 'bad'

model performance; thus, having implications for their different applications.

### 1.5 Objectives

In this study, the main objective is to assess the ability of different GHMs freely available from collaborators within the Global Flood Partnership (GFP), identifying which approaches are most suitable in different areas of the Amazon basin for simulating flood peaks. To pursue this objective, the analysis is designed to answer the following research questions:

1. How well do GHMs represent the annual hydrological regime in terms of the Kling Gupta Efficiency (KGE) and its individual components?
   2. Which model set-up best represents annual maximum river flows?
   3. Which hydrological routing model allows the best representation of daily and peak river flows?
   4. Which precipitation dataset allows the best representation of daily and peak river flows?
5. How do results differ when using a LSM as opposed to a hydrological model?
   6. By how much does calibration of groundwater and routing model parameters improve performance?

### 2 Data and methodology

The experimental design involves comparing the output of daily and annual maximum discharge estimates produced by different GHMs forced using atmospheric reanalysis or satellite precipitation datasets against observations of streamflow.

The common validation period is 1997-2015 with results also analysed for the shorter period of 2004-2015 to account for the shorter record length of one simulation.

### 2.1 Observations

Observed daily discharge data is used to evaluate each of the model runs. The network of hydrometric gauges is controlled and maintained by the national institutions responsible for hydrological monitoring in countries situated within the Amazon

basin. These include: Agência Nacional de Águas (Water National Office – ANA, Brazil), Servicio Nacional de Meteorología e Hidrología (National Meteorology and Hydrology Service – SENAMHI, Peru and Bolivia), Instituto Nacional Meteorologia e Hidrologia (Institute to Meteorology and Hydrology, INAMHI, Ecuador) and the Instituto de Hidrología, Meteorología y Estudios Ambientales (Institute of Hydrology, Meteorology and Environmental Studies - IDEAM, Colombia).

Daily water level values are collected by the respective institution and are sourced through the ORE-HYBAM observational service (see http://www.ore-hybam.org/) or directly from the national services. A time series of daily river flow for each station is obtained using stage and rating curve measurements which were determined using an acoustic Doppler current profiler (ADCP) conducted by the ORE-HYBAM observatory and SENAMHI (Espinoza et al., 2014). In total 75 hydrological stations throughout the Amazon basin are selected with an average record length of 17 years within the main

validation period (1997-2015). The locations of stations and their characteristics are displayed in Fig. 1a and Table S1

respectively. Stations selected have a minimum of five consecutive years' worth of data during the main validation period. The threshold was set to five to prevent the elimination of stations in data scarce areas such as Peru, Bolivia and Colombia.

## 2.2 Routing models and meteorological datasets

Eight GHMs composed of different meteorological datasets, hydrological/LSMs and river routing models, are used to each simulate river discharge across the Amazon basin. Four meteorological products (ERA-Interim Land re-analysis, ERA-5 re-analysis, European Centre for Medium-range Weather Forecasts (ECMWF) 20-year control reforecasts (hereafter defined as reforecasts) and the real-time TRMM TMPA 3B42 v.7), three hydrological/LSMs (PCR-GLOBWB, the Hydrology-Tiled ECMWF Scheme for Surface Exchanges over Land; H-TESSEL, EF5) and three river routing models (Catchment-based Macro-scale Floodplain model; CaMa-Flood, Lisflood and the Coupled Routing and Excess Storage; CREST) are employed. While the focus of this study is on GHMs made available by the GFP community, other models are available within the Amazon basin. Some examples include: MGB-IPH (Paiva et al., 2013), LPJmL (Lund–Potsdam–Jena managed Land; Bondeau et al., 2007), WaterGAP (water - global analysis and prognosis; Döll et al., 2003) and MAC-PDM.09 (the Macro-scale-Probability-Distributed Moisture model.09; Gosling & Arnell, 2011).

As a result of using freely available datasets from collaborators within the GFP, simulations are composed of a combination of routing models and meteorological datasets and do not all use the same precipitation input or hydrological set-up. However, the available combinations allow enough insight into the model components to draw conclusions for the objectives stated. For example, to analyse the performance of precipitation inputs, ERA-Interim Land, ERA-5 and the reforecasts are forced through the calibrated version of Lisflood, whereby the routing and LSM remain consistent. To evaluate the differences between using the Lisflood and CaMa-Flood routing models, two simulations which use ERA-Interim Land precipitation and the LSM H-TESSEL are compared. To identify the differences between employing a hydrological (PCR-GLOBWB) or LSM (H-TESSEL), two set-ups which use the ERA-Interim Land precipitation reanalysis and the CaMa-Flood river routing model are directly compared. Finally, to see how much benefit model calibration within Lisflood provides, ERA-Interim Land and ERA-5 are forced through the calibrated and un-calibrated Lisflood model versions. The CREST EF5 run is the sole simulation to have a unique hydrological model and meteorological input and although it is more challenging to analyse the performance of specific components of the model set-up against other simulations, it was included in the analysis for completeness.

An alternative approach would be to implement a full intercomparison experiment and run a new set of simulations which included all combinations of precipitation input, GHM and routing scheme. However, this is a very large undertaking and the time and computational expense to achieve this is prohibitive. Instead, by using freely available datasets with different hydrological set-ups, our method allows a first analysis providing enough evidence of dataset reliability and accuracy in order to determine the utility of the differing approaches for climate studies and to forecast applications. Moreover, by using iterative runs of similar model set-ups (i.e. changing a specific part of the hydrological model chain) it allows us to make

conclusive statements regarding the differences in skill. Finally, a short description of each model and atmospheric product is outlined below with a summary of each simulation provided in Table 1.

### 2.2.1 Precipitation datasets

**ERA-Interim Land** is a global reanalysis of land surface parameters produced by the ECMWF with a T255 spectral
resolution (~80 km or ~$0.75^0$; Balsamo et al., 2015). ERA-Interim Land was produced using the latest version of the land surface H-TESSEL model using atmospheric forcing from ERA-Interim (Dee et al., 2011), with precipitation adjustments based on the Global Precipitation Climate Project (GPCP) v2.1. Precipitation improvements were achieved by Balsamo et al. (2010) using a scale-selective rescaling procedure in which ERA-Interim 3-hourly precipitation were corrected to match the monthly accumulation provided by the GPCP at grid point scale (Huffman et al., 2009). All simulations which use ERA-
Interim Land are run offline to force the associated rainfall-runoff models (see Table 1). For a detailed description of the ERA-Interim Land and ERA-Interim datasets see Balsamo et al. (2015) and Dee et al. (2011) respectively. Dataset available at: <http://apps.ecmwf.int/datasets/data/interim-full-daily/levtype=sfc/>.

**ERA-5** is the latest reanalysis product of the ECMWF producing consistent estimates of atmospheric, land and ocean variables at a horizontal resolution of ~31 km, while the vertical atmosphere is discretised into 137 levels to 0.01 hPa
(ECMWF, 2018). ERA-5 is based on the Integrated Forecasting System (IFS) Cycle 41r2 which was used operationally at the ECMWF in 2016. Early analysis has shown that ERA-5 has an improved representation of precipitation (particularly over land in the deep tropics), evaporation and soil moisture compared to its predecessor ERA-Interim Land (ECMWF, 2017). ERA-5 is currently being produced in three "streams" and will eventually cover the period 1950 to near real-time (~3 days) with its completion due in 2019 (Emerton et al., 2018). Dataset available at:
<https://software.ecmwf.int/wiki/display/CKB/How+to+download+ERA5+data+via+the+ECMWF+Web+API>.

**ECMWF reforecasts** are a collection of historical forecasts from start dates at the same day of the year going back for a specific number of years to provide a consistent model climatology from which to compare forecasts (ECMWF, 2016). In this study we use the control member of the reforecasts which are created based on a retrospective run of the most recent version of the ECMWF's IFS to provide surface and subsurface runoff as input to the Lisflood routing model at a resolution
of $0.1^0$. The reforecast run is computed using a lighter configuration (11 ensemble members, run twice a week on Mondays and Thursdays) to reduce computational time. The purpose of running the ECMWF forecasts through the Lisflood routing model is to generate a long term (20-year) dataset which is consistent with operational GloFAS forecasts enabling the suitability of the dataset for use in the calibration of the Lisflood model parameters (Hirpa et al., 2018). This data covers the period June 1995 to June 2015 and due to frequent model updates of the IFS, is based on multiple model cycles: Cycle 41r1
(July through March) and Cycle 41r2 (March through June). The control reforecasts from Mondays and Thursdays are used subsequently to fill the whole weeks by taking the first 3- and 4-day forecast periods respectively throughout the 20 years.

**TRMM TMPA 3B42 RT v7** is a global merged multi-satellite precipitation product generated at the National Aeronautics and Space Administration (NASA). TMPA is computed for two products: a near real-time version (TMPA 3B42RT v7) and a post real-time gauged adjusted research version (TMPA 3B42 v7), both of which run at resolution of 3 hourly x $0.25^0$ x $0.25^0$ (Huffman et al., 2007). The TMPA 3B42 RT gridded dataset used in this study covers the global latitude belt from $60^0$

N to $60^0$ S. For further information see Huffman et al. (2007). Dataset available at: < https://pmm.nasa.gov/data-access/downloads/trmm>.

### 2.2.2 Hydrological and land surface models

**H-TESSEL** provides the land surface component of the ECMWF IFS (van den Hurk et al. 2000; van den Hurk and Viterbo 2003; Balsamo et al. 2009). H-TESSEL simulates the land surface response to atmospheric conditions estimating water and

energy fluxes (heat, moisture and momentum) on the land surface (Zsoter et al., 2019). H-TESSEL is predominately used within the operational set-up of short to seasonal-range weather forecasts coupled with the atmosphere, but it can also be used in an "offline mode" to calculate the land surface response to atmospheric forcing, whereby input data (e.g. near surface meteorological conditions) is provided on a 3 hourly timestep (Pappenberger et al., 2012). In this study, H-TESSEL receives boundary conditions from the atmospheric input provided by either the ERA-5 reanalysis, ERA-Interim Land reanalysis or

the reforecasts providing total runoff for the CaMa-Flood routing model, and the surface and sub-surface water fluxes for Lisflood. Runs forced using the ERA-Interim Land reanalysis are run in the offline mode. For a detailed description of H-TESSEL see Balsamo et al. (2009).

**PCR-GLOBWB** is a global hydrological and water resource model developed at the Department of Physical Geography, Utrecht University, Netherlands (Sutanudjaja et al., 2018). For each grid cell and time step, PCR-GLOBWB simulates

moisture storage in two vertically stacked upper soil layers, as well as the water exchange among the soil, the atmosphere, and the underlying groundwater reservoir. Besides, water demands for irrigation, livestock, industry, and households can be integrated within the model. Run-off is routed along a Local Drainage Direction (LDD) network using the kinematic routing wave equation. PCR-GLOBWB was applied at a resolution of 30 arcmin (~ 55 km x 55 km at the Equator) with meteorological forcing provided from the ERA-Interim Land reanalysis dataset between 1997 and 2015. For further

information on PCR-GLOBWB, see van Beek and Bierkens (2008), van Beek et al. (2011) and Sutanudjaja et al. (2018).

**EF5** is an open source software package developed at the University of Oklahoma (OU) that consists of multiple hydrological model cores producing outputs of streamflow, water depth and soil moisture (Clark et al., 2016). Since 2016, EF5 has been used operationally for local forecasts across the U.S. National Weather Service (NWS) for flash flooding purposes (Gourely et al., 2017). EF5 incorporates CREST which is a distributed hydrological model created by OU and

NASA (Wang et al., 2011). Within CREST, runoff generation, evapotranspiration, infiltration and surface and subsurface routing are computed at each grid cell within the model domain with surface and subsurface water routed using a kinematic wave assumption. Four excess storage reservoirs characterise the vertical profile within a cell representing interception by

the vegetation canopy and subsurface water storage in the three soil layers (Meng et al., 2013). In addition, the representation of sub-grid cell routing and soil moisture variability is made through the use of two linear reservoirs for overland and subsurface runoff individually (Wang et al., 2011). Locations of major streams, flow direction maps and flow accumulation are all derived from the HydroSHEDS (Hydrological Data and Maps Based on Shuttle Elevation Derivatives at Multiple

Scales) dataset (Lenhnar et al., 2008).

In this study, an un-calibrated version of EF5 was run using CREST version 2.0 (Xue et al., 2013; Zhang et al., 2015) for 13 years (2003-2015), with a one-year spin-up at a spatial resolution of $0.05^0$ x $0.05^0$. Parameters are estimated a priori from soil and geomorphological variables with meteorological forcing provided by the TMPA 3B42 RT product for precipitation and monthly averaged potential evapotranspiration (PET) from the Food and Agriculture Organisation (FAO). For full details on

the system set-up see Clark et al. (2016).

### 2.2.3 Routing models

**Lisflood** is a global spatially distributed, grid based hydrological and channel routing model commonly used for the simulation of large-scale river basins (van Der Knijff et al., 2010). It is currently used as an operational rainfall-runoff model within the European Flood Awareness System (EFAS) for streamflow forecasts over Europe (Smith et al., 2016). Unlike

EFAS, which uses the full Lisflood set-up, GloFAS and the simulations included in this study use only the routing component of the Lisflood set-up with surface and sub-surface input fluxes (e.g. vertical water, water/snow storage) provided by the H-TESSEL module of the IFS at a resolution of $0.1^0$. Surface runoff is routed through Lisflood using a four-point implicit finite-difference solution of the kinematic equations. Sub-surface storage and transport is routed to the nearest downstream channel pixel within one-time step through two linear reservoirs (Alfieri et al., 2013). The water in each channel

pixel is finally routed through the river network taken from the HydroSHEDS project (Lenhnar et al., 2008) using the same kinematic wave equations as for the overland flow. Subsurface flow from the upper and lower groundwater zones is routed into the nearest downstream channel as a scaled sum of the total outflow from both the upper and lower groundwater zones.

Lisflood also represents lakes and reservoirs as simulated points on the river network (Zajac et al., 2017). The outflow of lakes and reservoirs are based on: (a) upstream inflow, (b) precipitation over the lake or reservoir, (c) evaporation from the

lake or reservoir, (d) the lakes initial level, (e) lake outlet characteristics and (f) reservoir-specific characteristics. For further details on the parameterisation of lakes and reservoirs within Lisflood see Appendix A within Zajac et al. (2017). In the Amazon, represented lakes are predominately located along the main stem with very few reservoirs throughout the basin. For exact lake and reservoir locations within the global Lisflood model see Zajac et al. (2017).

In this study, two set-ups of Lisflood are used (Lisflood_uc and Lisflood_c). Lisflood_c represents the calibrated set-up of

the Lisflood routing and groundwater parameters (see Hirpa et al., 2018), while Lisflood_uc represents the uncalibrated model run. Parameters were calibrated with the reforecasts initialised with the ERA-Interim land reanalysis from 1995-2015

as forcing, against observed discharge data at 1278 gauging stations worldwide. All but one station (*40*, see Fig. 1a & Table S1) used in this study were included within the calibration. An evolutionary optimization algorithm was used to perform the calibration with the KGE used as the objective function. The calibration was carried out for parameters controlling the time constants in the upper and lower zones, percolation rate, groundwater loss, channel Manning's coefficient, the lake outflow width, the balance between normal and flood storage of a reservoir and the multiplier used to adjust the magnitude of the normal outflow from a reservoir. The results were validated by Hirpa et al. (2018) using the KGE (Gupta et al., 2009) over the period 1995-2015. In calibration (validation) KGE skill scores were greater than 0.08 compared to the default Lisflood simulation for 67% (60%) of stations globally. For a detailed description of the calibration of the Lisflood parameters and the range of values used for each parameter see Hirpa et al. (2018). Further details of the Lisflood model is described in van der Knijff et al. (2010).

**CaMa-Flood** is a global distributed river routing model which is forced by runoff input from a LSM or hydrological model to simulate water storage where further hydrological variables (i.e. river flow, water level and inundated area) can be derived along a prescribed river network. Horizontal water transport along the river network is calculated using the local inertia equations (Yamazaki et al., 2011). The backwater effect (i.e. upstream water levels which affect flow velocity downstream, see Meade et al., 1991) is represented by estimating flow velocity based on water slope (Yamazaki et al., 2011). Moreover, floodplain inundation is represented within CaMa-Flood as a subgrid scale process by discretising the river basin into unit catchments which consist of subgrid river and floodplain topography parameters (Yamazaki et al., 2014b). These parameters describe the relationship between the total water storage in each grid point and water stage and are automatically generated using the Flexible Location of Waterways (FLOW) method with the generation of the river map created by upscaling the HydroSHEDS flow direction map (Lehner et al., 2008). For further information about the CaMa-Flood model see the aforementioned references. In this study, daily river discharge was obtained using CaMa-Flood version 3.6.1 at a spatial resolution of $0.25^0$ (~25 km grid size) for both runs. The Manning's river and floodplain roughness coefficients were set at 0.03 s $m^{-1/3}$ and 0.10 s $m^{-1/3}$ uniformly for both CaMa-Flood simulations.

### 2.3 Verification metrics

### 2.3.1 Spearman's ranked correlation

The non-parametric Spearman's rho is used to measure the strength and direction of the monotonic relationship between the ranks of the observed and simulated annual maximum values. The non-parametric Spearman's rho was preferred to the Pearson's statistic as non-parametric measures are less sensitive to outliers in the data and are widely considered a more robust measure of the correlation between observed and predicted values (Legates & McCabe, 1999). Correlation scores for rho range from -1 to 1, with 1 being a perfect correlation. We consider scores which have a value of 0.6 or more to be considered skilful. Similar scores (between 0.5-0.7) are considered to represent a good level of agreement between observed and simulated values in similar studies (see Yamazaki et al., 2012; Alfieri et al., 2013).

### 2.3.2 KGE

The KGE (Gupta et al., 2009) measures the goodness-of-fit between estimates of simulated discharge and gauged observations and is a modified version of the dimensionless Nash Sutcliffe Efficiency (NSE; Nash and Sutcliffe, 1970). The metric decomposes the NSE into three independent hydrograph components (linear correlation (r), bias ratio (β) and relative variability between the observed and simulated streamflow (α)) by re-weighting the relative importance of each (Revilla-Romero et al., 2015). KGE values range from - ∞ to 1 with values closer to 1 indicating better model performance. To provide further context to the computed KGE scores, we use the breakdown of KGE values into four benchmark categories as according to (Kling, 2012). These are classified as follows:

- "Good" (KGE $\geqslant$ 0.75)
- "Intermediate" (0.75 > KGE $\geqslant$ 0.5)
- "Poor" (0.5 > KGE > 0)
- "Very poor" (KGE $\leqslant$ 0)

Although originally for the modified version of the KGE, these categories provide an informative benchmark at which to evaluate results. A similar study (Thiemig et al., 2013) assessing the performance of satellite-based precipitation products for hydrological evaluation also adopted the same approach.

When analysing the results, each component of the KGE is also considered independently enabling model errors to be directly related to either the variability (KGE_ α), bias ratio (KGE_ β) or correlation (KGE_r; Guse et al., 2017). KGE_α values greater than 1 indicates that variability in the simulated time series is higher than that of the observed. Values less than 1 show the opposite effect. KGE_ β values greater than 1 indicate a positive bias whereby predictions overestimate flows relative to the observed data, while values less than 1 represent an underestimation.

To evaluate the relative improvement of using one model set-up relative to another (e.g. using the calibrated Lisflood routing model as opposed to the uncalibrated model version) metrics are calculated as skill scores:

$$KGE_{SS} = \frac{KGE_a - KGE_{def}}{1 - KGE_{def}} \qquad\qquad (1)$$

Where: $KGE_{SS}$ signifies the KGE skill score, $KGE_a$ is the KGE score for the improved run or simulation of interest (e.g. Lisflood_c) and $KGE_{def}$ is the KGE score for the 'default' or comparative run (e.g. Lisflood_uc). Positive $KGE_{ss}$ indicates improved skill whilst a negative score represents a decrease in skill. For each case, KGE scores are calculated against observed river flow data. The correlation skill score is calculated similarly. All metrics are computed in the R environment using the 'verification' (Gilleland, 2015) and 'hydroGOF' (Zambrano-Bigiarini, 2017) R-packages.

# 3 Results and discussion

To allow for easier interpretation, the results and discussion are separated into six sections which match the research questions presented in Sect. 1.5, in addition to an outline of potential future work. Due to similar results between the two validation periods (1997-2015 and 2004-2015), only results for 1997-2015 are shown. For 2004-2015 results see Figs. S1 & S2. Results and discussions for individual stations are commonly referred to by the station numbers in italics and are presented in Fig. 1a and Table S1.

## 3.1 How well is the annual hydrological regime represented?

The annual hydrological regime on average is well represented by all models (Fig. 2), with the rationale for poorer performance at specific gauges dependent on either the temporal correlation, bias ratio or variability ratio components of the KGE (Figs. 3-5). An average of 50% of stations note scores above 0.5 for the KGE metric across all eight simulated runs with a maximum value of 0.92 observed at the Santa Rosa gauging site (*48*, Fig. 1a) for the ERA-5 Lisflood_c simulation (Fig. 2f). The two CaMa-Flood set-ups using the hydrological model PCR-GLOBWB and the LSM H-TESSEL show the lowest skill with 19 and 18 stations noting scores greater than 0.5 respectively. On the contrary, best performance is from the calibrated Lisflood set-ups with median scores across stations of 0.56, 0.63 and 0.64 for runs forced with ERA-Interim Land, the reforecasts and ERA-5 respectively. Such results are unsurprising given that the KGE was used as the objective function in the calibration algorithm of the Lisflood routing model.

In terms of spatial distribution, poorest performance is consistent for the majority of simulations at the Arapari (*55*), Boca Do Inferno (*56*) and Base Alalau (*61*) gauging stations located north of Manaus, at the Fazenda Cajupiranga gauge (*64*) in the northernmost Branco catchment and at the Fontanilhas (*35*) and Indeco (*49*) stations in the south-eastern Brazilian Amazon (Fig. 2). In the south-eastern Amazon, particularly in the Madeira and Tapajos sub-basins, the quantity of existing or under construction dams is at its highest (Fig. 1b). Damming of rivers is known to have impacts on different aspects of the flow regime with possible alterations in the timing, magnitude and frequency of low and high flows (Magilligan & Nislow, 2005). Indeed, the frequency and duration of low and high flow pulses at stations downstream of dams has been shown to be particularly affected by the construction of cumulative dams (Timpe & Kaplan, 2017). Thus, discrepancies between observed modelled data shown in Fig. 2 could be due to alterations to key features of the flow regime.

Highest scoring stations (KGE score > 0.75) are predominately found in the south-western Brazilian Amazon where the network of tributaries remain relatively unaffected by damming and where slopes are gentle (Figs. 1b, d). However, high skill at stations (*32*, *33* & *43*) along the Madeira river for most simulations (Fig. 2) highlight that the impacts of hydroelectric dams needs to be considered on an individual basis with two of the largest dams (> 3000 MW) situated along the river (see Fig. 1b).

Figures 3, 4 and 5 show the breakdown of the KGE scores for each hydrological component to evaluate differences in performance with respect to the correlation (i.e. timing), flow variability ($\alpha$) and bias ratio ($\beta$). An average of 79% of stations note correlation coefficients exceeding 0.6 across all runs with those using the Lisflood routing model performing similarly in both spatial distribution and magnitude (Fig. 3). In contrast, 51% and 47% of stations achieve values exceeding 0.6 for

CaMa-Flood H-TESSEL and CaMa-Flood PCR-GLOBWB respectively, with the hydrological model, PCR-GLOBWB noting better performance at stations along the main-stem. The increased performance of Lisflood relative to simulations incorporating CaMa-Flood are likely due to the increased spatial resolution of the routing component (see Table 1). This is supported by results for CREST EF5, with 76% of stations noting values above 0.6 and the model occupying a finer spatial resolution than that of CaMa-Flood (Fig. 3g).

The variance of modelled river flow is on average higher than the observed time series in all of the simulations with the exception of the ERA-Interim Land PCR-GLOBWB CaMa-Flood simulation. For this run, 85% of stations observe values of less than one with stations situated in the Peruvian Amazon (*2*, *3*, *4* and *5*) the notable exception (Fig. 4b). In contrast, 79% of stations for the CaMa-Flood set-up using the LSM H-TESSEL, note values greater than one (Fig. 4a). All runs tend to underestimate river flows relative to the observed time series with the majority of stations observing a beta value of less than

one (Fig. 5). In the calibrated Lisflood simulation forced with the reforecasts, almost half of all stations observe scores between 0.9 and 1.1 (i.e. grey circles), with a median of 0.99 (Table 2). These results are not replicated in the other two calibrated runs when using either ERA-Interim Land or ERA-5 as the precipitation input (Figs. 5d & 5f). For both of these runs a decrease is found in the number of stations achieving scores between 0.9 and 1.1 relative to the associated uncalibrated Lisflood set-ups (Figs. 5c & 5e). This is also highlighted by a decrease in the median scores of the two

respected runs (Table 2), meaning that a greater water deficit exists in the calibrated set-ups.

Stations in the south-eastern Amazon, particularly in the upper reaches of the Teles Pires river (*37*, *38* & *49*), tend to underestimate river flow for most simulations (Fig. 5). In this region of the basin precipitation is controlled by frontal systems in the South Atlantic Convergence Zone (SACZ), which is prevalent during austral summer (Ronchail et al., 2002; Espinoza et al., 2009). In addition, rainfall variability in the Amazon is strongest in the south-east with a distinct dry season

(Paiva et al., 2012; Espinoza et al., 2009). Further analysis could be useful in evaluating seasonal patterns of model performance to establish whether climatological features such as the SACZ are accurately represented within the precipitation datasets. Other factors impacting performance in the south east could be associated with the geology and topography (Figs. 1c, d). Stations in this area of the basin are located within the Brazilian Shields, composed predominately of Precambrian rock and are characterised by gentle slopes and low erosion rates (Filizola and Guyot, 2009). Paiva et al.

(2012) demonstrated the importance of accurate initial conditions of groundwater state variables in Tapajos and Xingu river basins, particularly for low flows. In comparison, the majority of the central parts of the basin are characterised by tertiary rocks, flat terrain, large floodplains and high sediment yields. In these regions (e.g. in the south-western Brazilian Amazon),

KGE scores are generally higher (Fig. 2), with surface water variables (e.g. water levels, surface runoff and floodplain storage) considered more important in hydrological prediction uncertainties (Paiva et al., 2012).

The KGE allows us to make explicit interpretations into the hydrological performance of each model owing to decomposition into correlation, bias and variability terms (Kling et al., 2012). The results indicate that the required developments to improve the representation of daily river flows is specific to each individual model and to the area of interest. For instance, for the ERA-Interim Land PCR-GLOBWB run, daily correlation scores (Fig. 3b) showed the model suffers at reproducing the temporal dynamics of flow (as measured by r) in northern catchments. Calibration of parameters which control the timing of the flood wave (e.g. river flow velocity) may improve performance. Whereas, model set-ups incorporating the uncalibrated Lisflood routing model generally had lower KGE values in the east of the basin corresponding to an overestimation of river flow variability (Figs. 4c, e). For these runs, performance slightly improved upon the calibration of the groundwater and routing parameters relating to timing, flow variability and groundwater loss (Figs. 4d, f).

### 3.2 Which model set-up best represents annual maximum river flows?

Both the calibrated and uncalibrated versions of Lisflood simulations forced with the ERA-5 reanalysis are the best performing runs with median scores of 0.53 and 0.54 for the uncalibrated and calibrated simulations respectively (Fig. 7 & Table 2). However, a large deterioration in skill is evident for all simulations for Spearman's ranked coefficients between observed and predicted annual maximum river flows (Fig. 6) with only 21% of stations on average observing scores exceeding 0.6 across all simulations. Here, it is important to note that due to the length of some station time series the number of overlapping data points can be small and therefore the spatial distribution of model performance should be interpreted with caution. To provide a certain level of confidence between results, stations whose time series equals or exceeds 15 years are denoted using a circle, whereas those between 10-14 and 5-9 are represented using a square and triangle respectively.

Highest scores are generally located towards the eastern side of the basin and along the main Amazon River where the terrain is predominately flat, and rivers drain extensive floodplains. These are constrained to runs using the Lisflood routing model with either ERA-Interim Land or ERA-5 as forcing (Figs. 6c-f). Interestingly, the calibrated Lisflood set-up forced using the reforecasts does not replicate good performance in these regions (Fig. 6h), indicating that the error between simulated and observed peak river flows could be associated with the precipitation input. When observing daily mean precipitation totals over the validation period (1997-2015), the reforecasts observe lower precipitation totals over central to northern areas of the basin relative to both of the climate reanalysis datasets (Fig. 8). However, when comparing the results of ERA-Interim Land H-TESSEL CaMa-Flood and the ERA-Interim Land H-TESSEL Lisflood_uc set-ups, correlations are much lower in the CaMa-Flood simulation, suggesting that both precipitation and routing processes are equally important (Figs. 6a & 6c).

Low agreement between peaks is consistent in the south-east and north-west of the basin across all simulations (Fig. 6). In the south-east, a lack of skill could again be associated with the abundance of hydroelectric dams in the region or through the poor representation of the SACZ rainfall regime. Evaluating the ability to represent the timing and magnitude of the annual flood wave has important implications for models predicting flood hazard and for practices providing early warning
information. These results identify that while the representation of daily river flows improves upon model calibration of the Lisflood routing model (Sect. 3.1), the influence of routing calibration for simulating flood peaks has no impact.

### 3.3 What is the best performing hydrological routing model?

We assessed the performance of the CaMa-Flood and Lisflood_uc routing models by comparing the two runs which are forced using the ERA-Interim Land reanalysis dataset. On average the uncalibrated Lisflood run outperforms CaMa-Flood
for all metrics analysed (Fig. 7 & Table 2). Results from the CREST EF5 model are also discussed but are not directly comparable due to using differing meteorological inputs.

The median score of the correlation component of the KGE (i.e. Pearson's correlation coefficient) is found to increase by 0.19 when using the un-calibrated Lisflood model relative to CaMa-Flood with 28 more stations achieving a correlation score of 0.6 or higher (Figs. 3a & c). This number increases when considering correlation scores greater than 0.8 with 38 and
seven stations reaching this value for Lisflood and CaMa-Flood respectively. The most notable increase in skill is found in Peru along the Marañón and Napo rivers (*2 & 5*), which note an increase of 0.85 and 0.71 respectively when using the Lisflood model. In comparison, the CREST EF5 simulation fits between the CaMa-Flood and Lisflood runs with a median daily correlation score of 0.71 and notes 12 stations which have scores greater than 0.8 (Fig. 3g).

For the overall KGE metric, 24% and 3% of stations have values exceeding 0.5 and 0.75 for CaMa-Flood. These figures rise
to 52% and 11% respectively in the uncalibrated Lisflood run. Large differences are particularly notable at stations situated in the upper reaches of the Solimões River (*2-6*) and within a cluster of stations situated towards the Colombian Amazon in the north-west (Fig. 2c). Significant differences are identified for peak flow correlations with only three stations (*27, 17* and *22*) achieving scores exceeding 0.6 for the CaMa-Flood simulation compared to 22 using the uncalibrated Lisflood routing scheme (Figs. 6a & 6c). In comparison, the CREST EF5 simulation has 11 stations exceeding this threshold with no
distinguishable spatial pattern (Fig. 6g). For this run, the time series of modelled data is shorter (2004-2015) and so peak flow correlations should be interpreted with caution.

Stations located in and around the main Amazon River observe better performance for representing flood peaks in the Lisflood simulation (Fig. 6c), aligning with the locations of lakes included within the Lisflood set-up (see Zajac et al., 2017). This level of skill was not replicated in the CaMa-Flood simulation where the representation of lakes is not included (Fig.
6a), suggesting the potential importance of lake parameterisation for accurate peak flow estimations. However, Zajac et al. (2017) demonstrated that although the inclusion of lakes in Lisflood was found to generally improve the representation of

extreme discharge for the five and twenty year return periods on the global domain, the change in skill upon the inclusion of lakes and reservoirs in the Amazon was minimal for several metrics. Very few reservoirs are included within Lisflood in the Amazon and therefore the estimated effects on simulated streamflow is restricted.

Zhao et al. (2017) concluded the importance in the choice of different river routing schemes for simulating peak discharge across the globe. While Hoch et al. (2017b) comparison of two routing models found results to differ despite having identical boundary conditions. It is therefore of interest to evaluate not only the entire GHM set-up but also to assess the suitability of each model component of the hydrological chain in order to determine which routing model is most suitable for certain applications within the Amazon basin. Results suggests that adjustments of certain parameters such as the Manning's channel coefficient could potentially improve the performance of the CaMa-Flood model, with the default coefficient higher in the uncalibrated Lisflood set-up (0.10 as opposed to 0.03; see Hirpa et al., 2018 for all default parameter values).

### 3.4 What is the best performing precipitation dataset?

Three precipitation products (ERA-Interim Land, ERA-5 and the reforecasts) are used to force the calibrated Lisflood routing model with the most recent ERA-5 reanalysis product the best performing dataset. Figure 8 displays mean daily precipitation totals for each dataset over the main validation period (1997-2015). Main differences can be seen in the far west of the basin towards the Andes mountains, where precipitation is higher in ERA-5 compared to ERA-Interim Land and in the north-west where average daily precipitation totals are smaller in the reforecasts. On the other hand, values in the south-eastern corner of the basin are very similar between the three datasets. When comparing observed and simulated annual peak flows, median correlation scores improve by 0.12 and 0.22 when using ERA-5 compared to using ERA-Interim Land and the reforecasts respectively (Table 2). 28 stations reach the 0.6 threshold relative to 22 and nine stations for ERA-Interim Land and the reforecasts respectively with the range of coefficients smaller for ERA-5 (Fig. 7a).

Figures 9e and 9f highlight the relative gain or loss in skill when using ERA-5 compared to ERA-Interim Land. Greatest improvements for each metric are observed within the upstream reaches of the Solimões River, particularly for stations located within the Peruvian Amazon (*2, 4 & 5*). In the main western headwater to the Solimões River (the Marañón river) at the San Regis gauging site (*2*) and at Tamshiyacu (*4*) near to the city of Iquitos, annual maximum correlation skill scores are 0.51 and 0.59 respectively. These results highlight that poor performance found in upstream reaches of the Solimões River (Fig. 6c & 6d) is likely due to the representation of rainfall rather than routing performance.

In the other main tributary to the Solimões River, the Ucayali river, simulated annual peak flows show little agreement with observed data with a decrease in skill identified when using ERA-5 as opposed to ERA-Interim Land (Fig. 9e). Despite the lack of agreement between observed and modelled data in the Ucayali river, the higher correlation scores identified downstream at Tamshiyacu suggests that better representation of high-water periods at the start of the Solimões River is

likely modulated by the larger Marañón river. Therefore, the ability to represent flood hazard in communities near to the city of Iquitos is more dependent on how well we can predict river flow in the Marañón river.

All three runs perform well for the KGE metric with little difference in results spatially (Figs. 2d, f, h). The reforecast simulation used within the Lisflood calibration is found to be superior with 75% of stations achieving scores which exceed
0.5 relative to 71% and 59% for ERA-5 and ERA-Interim Land respectively. Increased skill in the Peruvian Amazon is again the most noteworthy (Fig. 9f) with KGE skill scores of 0.67 for the Requena (*3*) (Ucayali river) and San Regis (*2*) (Marañón river) stations and 0.71 for Tamshiyacu (*4*) (Solimões River) when using ERA-5 relative to ERA-Interim Land. This increase in KGE skill can be attributed to an improvement in the variability and bias ratios found between the simulated and observed time series. Daily correlation scores for the three stations (*2, 3 & 4*) are near identical with the variance and bias ratios
underestimated for ERA-Interim Land, while being much closer to the observed data for ERA-5 (Figs. 4d, f & 5d, f).

The Tamshiyacu gauging station (*4*) is used to measure flood hazard in the city of Iquitos at the start of the Solimões River (Espinoza et al., 2013) and is therefore of particular interest. At this important location, scatterplots of observed against simulated river discharge (Fig. 10) show that the negative bias observed when using ERA-Interim Land is corrected for when using ERA-5 with the magnitude of the 90$^{th}$ percentile of river flows almost identical to that of the observed dataset.
Improvement is likely associated with the increased resolution of the ERA-5 reanalysis, which observes higher daily mean precipitation totals in regions towards the Andes in the far north west of the basin (Fig. 8b). Waters found at Tamshiyacu are of Andean origin meaning that the representation of rainfall in the Andes Mountains is fundamental to accurately predicting streamflow. ERA-5 runs at a horizontal resolution of ~31 km and includes an additional 73 vertical levels to 0.01 hPa compared to ERA-Interim Land, meaning the representation of the troposphere is enhanced (ECMWF, 2017).

The success of GHMs in producing adequate estimates of river flow is underpinned by uncertainties within the meteorological input (Butts et al., 2004; Beven, 2012; Sood & Smakhtin, 2015). These results have particular importance for flood forecasting applications and research concerning extreme floods with the higher resolution ERA-5 dataset providing closer agreement between observed and simulated annual maximum river flows, particularly for the Peruvian Amazon. With the time series of observed data often beginning in the 1980's in the Amazon, ERA-5 could provide a useful tool for
analysing historical flows and establishing links to climate variability. Upon completion, ERA-5 will date back to 1950 (Zsoter et al., 2019) meaning locations in which model skill is considered high could benefit from up to 30 years' worth of additional data for use in climate studies; thus, allowing for more robust analysis. In future work, it could be of interest to compare the performance of ERA-5 against a wider range of precipitation data sets, such as the Multi-Source Weighted-Ensemble Precipitation (MSWEP) product that carefully integrates gauge, satellite and reanalysis-based estimates. Beck et
al. (2017) evaluation of 22 precipitation datasets previously demonstrated the advantages of using merged products for hydrological modelling purposes.

### 3.5 How do results differ between using a LSM and a hydrological model?

The LSM H-TESSEL and the hydrological model PCR-GLOBWB are directly compared whereby the precipitation forcing (ERA-Interim Land) and river routing scheme (CaMa-Flood) are consistent. Overall, it appears that the choice between using a LSM or a hydrological model in the Amazon basin is dependent not only on the specific region of interest but also on the application and needs of the user. Previous studies (Zhang et al., 2016; Beck et al., 2017) have found that LSM models, on average, perform better in rainfall dominant regions, whereas hydrological models tend to achieve better results in snow dominated regions owing to the use of complex energy balance equations introducing additional uncertainties. For the Amazon basin, Spearman's rank correlation coefficients between simulated and observed peak river flow are closely matched with a median of 0.24 and 0.23 for H-TESSEL and PCR-GLOBWB respectively (Table 2). However, the number of stations with Spearman's maximum correlation scores exceeding 0.6 is slightly higher in PCR-GLOBWB at seven compared to three with H-TESSEL (Figs 6a & 6b).

To illustrate the gain or loss in skill when using H-TESSEL relative to PCR-GLOBWB, the Spearman's annual maximum correlation and KGE skill scores were calculated for each station (Figs. 9g & 9h). Overall, 68% of stations show improved skill for peak river flow correlations when using the LSM model, though the gain in skill is minimal (median correlation skill score = 0.06). This percentage drops to 37% and 22% for improvements in skill which exceeds 0.1 and 0.2 respectively (Fig. 9g). On the contrary, over half of stations see improvements in the KGE skill score for the hydrological model, PCR-GLOBWB and 23% of stations observe KGE skill score increases which exceed 0.25 (Fig. 9h).

A large loss in performance for the KGE is observed when using H-TESSEL for stations in the Peruvian Amazon at the confluence point to the Solimões River (Fig. 9h). Model performance in this region can largely be attributed to the failure of the H-TESSEL CaMa-Flood run to accurately represent the variance of flow and the temporal correlation component of the KGE with the variability of modelled flow far higher than in the observed data (Fig. 4a). Northern regions in the Branco basin and stations situated towards the Colombian Amazon show the opposite effect with higher KGE coefficients found for the H-TESSEL CaMa-Flood run (Fig. 2a), indicating that model suitability is regionally specific.

### 3.6 By how much does the calibration of groundwater and routing parameters improve performance?

Calibration of hydrological models is known to be a useful tool in providing more accurate estimates of river flow (Beck et al., 2017). However, due to a lack of data and the computational expense required in the calibration of GHMs, many remain uncalibrated (Bierkens, 2015; Sood & Smakhtin, 2015). Both Gupta et al. (2009) and Mizukami et al. (2019) demonstrate that square error type metrics are unsuitable for model calibration when the model in question requires robust performance for high river flows. Improvement of flow variability estimates was documented in both studies when switching the calibration metric from the NSE to the KGE for both a simple rainfall-runoff model (similar to the HBV model; Bergström, 1995) and for two more complex hydrological models (Variable Infiltration Capacity and mesoscale Hydrologic Model),

suggesting similar results are likely to be achieved for other hydrological models. To investigate the potential benefits of routing model calibration, whereby the KGE was used as the objective function, the time series of river discharge for the calibrated Lisflood runs forced using the ERA-Interim Land and ERA-5 reanalysis datasets were compared against the associated default set-ups without routing calibration.

Overall, hydrological performance improves upon model parameter calibration with positive KGE skill scores (i.e. an increase in skill) at 61% (59%) of gauging stations for simulations forced with ERA-Interim Land (ERA-5) (Figs. 9c & 9d). The influence of calibration is stronger for the simulation forced with ERA-5, with the number of stations achieving "intermediate" KGE scores (i.e. $0.75 > \text{KGE} \geqslant 0.5$) totalling 53 compared to 43 for ERA-Interim Land, an increase of nine and 12 stations relative to the associated uncalibrated runs. When observing the spatial distribution of relative improvements,

an east/west divide can be seen (Figs. 9c & 9d). Generally, decreases in skill are concentrated to stations in the western side of the basin, whereas stations located to the east display improved hydrological representation.

Three stations (*2, 3 & 4*) in the Peruvian Amazon show increased KGE skill scores when using the calibrated ERA-5 run relative to the similar uncalibrated set-up (Fig. 9d). Conversely, a loss in skill is observed at each station for the calibrated run forced using ERA-Interim Land (Fig. 9c). These results are likely associated to a larger negative runoff bias within the

ERA-Interim Land Lisflood_uc run relative to the ERA-5 Lisflood_uc simulation for the three stations (Figs. 5c and 5e). This is supported by Hirpa et al. (2018), who concluded that stations which have a negative streamflow bias in the default run (i.e. Lisflood_uc) also have a negative KGE skill score in the calibrated simulation owing to the challenge of correcting for a water deficit within the routing component. Thus, for GHMs which tend to underestimate runoff, adjustments of parameters within the LSM or hydrological model (e.g. those responsible for the portioning of precipitation into runoff) or

through bias correction measures within the precipitation dataset, may be advantageous in efforts to accurately represent floods.

No significant differences between calibrated and uncalibrated Lisflood annual maximum correlation scores are identified (Fig. 7a & Table 2). In total, the number of stations exceeding the 0.6 threshold for peak flow correlations remains the same for runs involving ERA-5 and decreases by one for ERA-Interim Land, meaning that the routing model calibration has very

little impact in the ability to capture annual peaks. This suggests that calibrated parameters controlling flow timing (e.g. Manning's channel coefficient) are not as important for simulating the magnitude of higher flows in the Amazon basin and that bias correction of the precipitation or calibration of parameters associated with runoff and evapotranspiration might be more useful. As previously highlighted by Hirpa et al. (2018), the inclusion of an objective function that is explicitly based on flood peaks could improve the ability of Lisflood to simulate floods. This is supported by previous studies (Greuell et al.,

2015; Beck et al., 2017; Mizukami et al., 2019) which have also identified that improved performance in calibrated models is predominately specific to metrics which are incorporated into the objective function used within the calibration. For instance, in Mizukami et al. (2019) they find that when using an application specific metric (Annual Peak Flow Bias; APFB) for the

calibration of two hydrological models, it produced the best peak flow annual estimates compared to using the NSE, KGE and its components. However, despite this improvement, flood magnitudes were still underestimated for all metrics used in calibration and the use of the APFB as the calibration metric resulted in poorer performance across the individual KGE components upon evaluation.

## 3.7 Limitations and future work

While estimating the magnitude of peak river flows is fundamental, more evaluation is required in assessing the ability to represent the timing of flood peaks. Modelled flood peaks have been known to occur too early in large Amazonian rivers (Alfieri et al., 2013; Hoch et al., 2017b) with accurate flow timing of significant importance in the Amazon basin. For example, the time displacement between peak flows in coinciding tributaries are known to play a major role in the dampening of the Amazon flood wave (Tomasella et al., 2010) and in the synchronisation of flood peaks, commonly associated with exceptional flood events (e.g. Marengo et al., 2012; Espinoza et al., 2013; Ovando et al., 2016). Additional evaluation using metrics which focus specifically on the timing aspect, such as the delay index (Paiva et al., 2013), would enable a more complete assessment of the hydrological modelling regime.

A limitation of this type of study is due to the intercomparison being restricted to the macroscale (i.e. only a subset of potential modelling configurations are considered). In future work it would be useful to increase the granularity of the modelling decision matrix to allow conclusions to be more generalised across the modelling community. For instance, when comparing the performance of the Lisflood and CaMa-Flood routing models, the results are specific to the simulations forced using the ERA-Interim Land reanalysis dataset. Although useful in providing a general indication of routing performance for each model when using a climate reanalysis dataset, the conclusions are specific to that particular comparison with differing results possible when using another precipitation input. Future work could investigate one of the research questions stated in the objectives (Sect 1.5) at a finer resolution. For example, by comparing several different runs which use the Lisflood and CaMa-Flood routing models, whereby a greater variety of precipitation inputs are considered (e.g. MSWEP, CHIRP V2.0, ERA-5, TRMM v.7 amongst others). Such analysis would allow more general conclusions and recommendations to be made to the modelling community who are interested in those particular routing schemes. A similar approach could be adopted for the assessment of other components of the hydrological modelling chain.

## 4 Conclusions

In this paper, eight different GHMs were employed in an intercomparison analysis using two verification metrics to assess model performance against gauged river discharge observations. The motivation for this work stemmed from the need to evaluate the ability of GHMs to reproduce historical floods in the Amazon basin for use in climate analysis and to identify the strengths and weaknesses which exist along the hydrological modelling chain in order to provide insight to model developers. The implications of these results suggest that the choice of precipitation dataset is the most influential

component of the GHM set-up in terms of our ability to recreate annual maximum river flows in the Amazon basin. This is evident with average station correlations between observed and simulated annual maximum river flows increasing when using the new ERA-5 reanalysis dataset, with significant improvements in locations of the Peruvian Amazon. In this region, waters are sourced from Andean origins where rainfall can often be poorly represented due to topographically complex

terrains (Paiva et al., 2013). Thus, those wishing to simulate higher flows in the upper reaches of the Amazon may benefit from choosing a precipitation dataset which has a high spatial resolution, whereby the upper atmosphere is discretised at finer scales. Although, an exact recommended spatial resolution cannot be provided based on the results of this study alone, previous works (e.g. Beck et al., 2017) support the need for a comparatively high-resolution data set in addition to other advantageous factors such as a long temporal record and the inclusion of daily gauge corrections.

Although parameter calibration of the Lisflood routing model improved the representation of the whole hydrological regime across the basin, the agreement between observed and simulated peak discharge values saw no change upon calibration. This indicates that the benefit of calibration is confined to the objective function used, in this case the KGE, and highlights that further model calibration using an objective function that fits the purpose of the application (e.g. RMSE of flood peaks or APFB for flood forecasting systems) could be worth considering. It is important to reiterate however, that thoughtful

consideration is required if choosing application specific metrics, with the potential to degrade performance in other aspects of the hydrological regime (e.g. bias and flow variability ratios) a concern (Mizukami et al., 2019). The relative importance of good performance in the specific target metric compared to better performance for a range of metrics should be assessed on a model by model and circumstantial basis, taking into account the needs of potential users.

*Data availability.* All of the data and models used in this study were obtained from collaborators of the Global Flood

Partnership (GFP) and are freely available. Access to these sources are mentioned in Sect. 2.

*Author contributions.* EZ provided data and information for all simulations incorporating Lisflood and for the ERA-Interim Land H-TESSEL CaMa-Flood set-up. ZF and JM provided data and information for the TRMM CREST EF5 and ERA-Interim Land PCR-GLOBWB CaMa-Flood runs respectively. ES, HC, JB and EC supervised the research and provided important advice. ES, HC and JT designed the analysis and JT undertook the research in addition to writing the paper. All

authors were involved in discussions throughout the development and commented on the manuscript.

*Competing interests.* The authors declare that they have no conflict of interest.

*Acknowledgements.* Jamie Towner is grateful for financial support from the Natural Environment Research Council (NERC) as part of the SCENARIO Doctoral Training Partnership (grant agreement NE/L002566/1). The first author is also grateful for additional travel support and funding provided by the Red Cross Red Crescent Climate Centre, to the observational and

national services, SO-HYBAM, SENAMHI, ANA and INAMHI for providing observed river discharge data and to the ECMWF for computer access and technical support. Finally, a specific thanks goes to Professor Christel Prudhomme and the

Environmental Forecasts team in the Evaluation Section at the ECMWF for their advice and support throughout the analysis and writing of the manuscript.

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

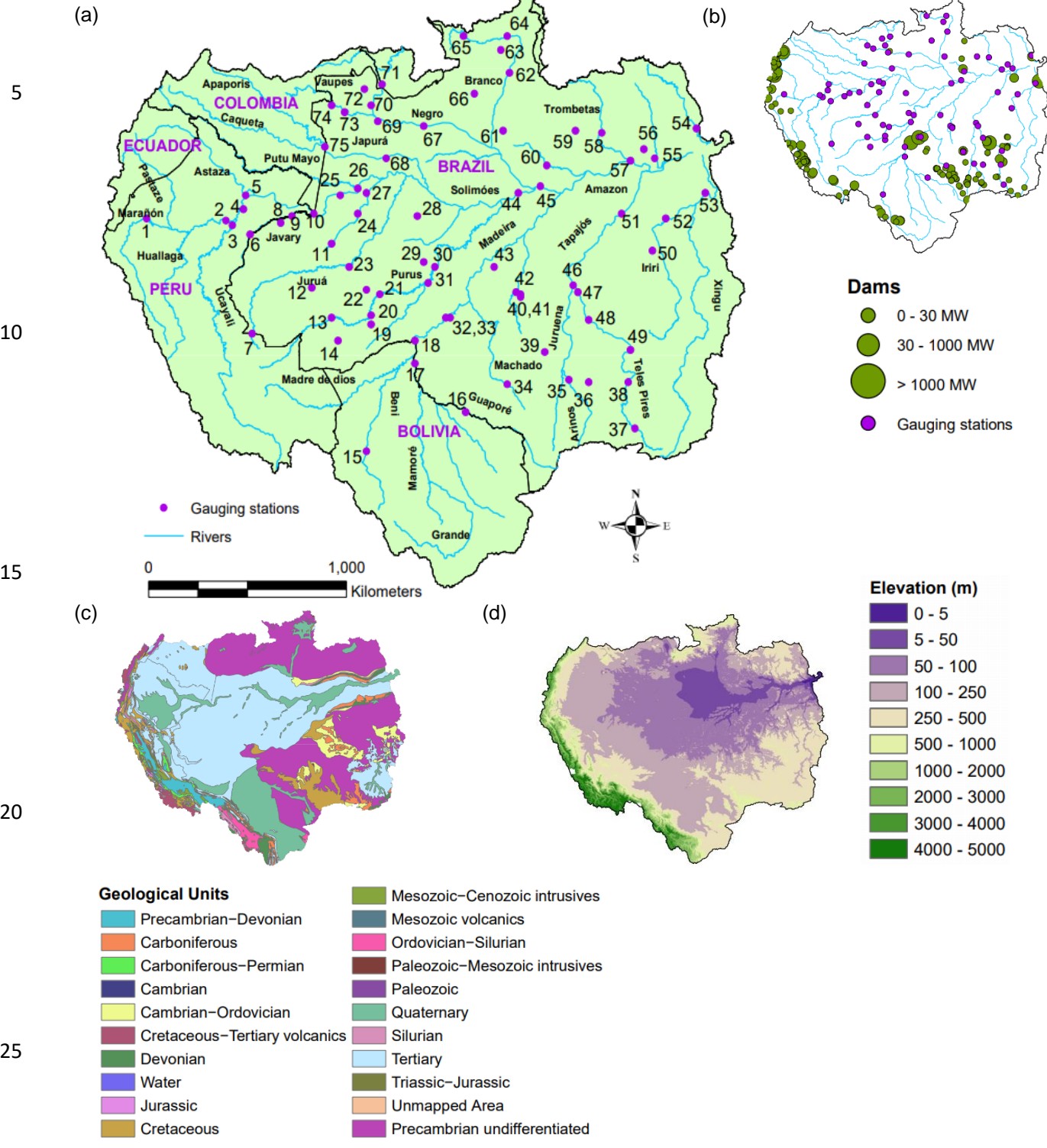

**Figure 1: (a) Locations of the 75 hydrological gauges and the river network of the Amazon basin. Numbers represent stations which are referred to throughout the main text in italics. For station information see Table S1. (b) Locations of existing and under-construction dams as of 2017 (see Latrubesse et al., 2017). (c) Geological map of the Amazon (Schenk et al., 1999). (d) Elevation map of the basin from the digital elevation model (DEM), GTOPO30, at a horizontal resolution of approximately 1 km (USGS, 1996)**

5   **Table 1: Characteristics of the eight Global Hydrological Models (GHMs) used to produce estimates of daily river discharge.**

| Model Run | Meteorological forcing[1] | GHM[2] | GHM Spatial Resolution | Routing Model[3] | Routing Spatial Resolution | Temporal Resolution | Start | End | Calibration | Authors |
|---|---|---|---|---|---|---|---|---|---|---|
| ERA-I Land H-TESSEL Lisflood_uc | ERA-I Land | H-TESSEL | ~$0.75^0$ (~80 km) | Lisflood | $0.10^0$ (~10 km) | Daily | 01 Jan 1997 | 31 Dec 2015 | None | Balsamo et al. (2015)[1] Balsamo et al. (2009)[2] van der Knijff et al. (2010)[3] |
| ERA-I Land H-TESSEL Lisflood_c | ERA-I Land | H-TESSEL | ~$0.75^0$ (~80 km) | Lisflood | $0.10^0$ (~10 km) | Daily | 01 Jan 1997 | 31 Dec 2015 | See Hirpa et al. (2018) | Balsamo et al. (2015)[1] Balsamo et al. (2009)[2] van der Knijff et al. (2010)[3] |
| ERA-5 H-TESSEL Lisflood_uc | ERA-5 | H-TESSEL | ~$0.28^0$ (~31 km) | Lisflood | $0.10^0$ (~10 km) | Daily | 01 Jan 1997 | 31 Dec 2015 | None | See ECMWF (2018)[1] Balsamo et al. (2009)[2] van der Knijff et al. (2010)[3] |
| ERA-5 Lisflood H-TESSEL_c | ERA-5 | H-TESSEL | ~$0.28^0$ (~31 km) | Lisflood | $0.10^0$ (~10 km) | Daily | 01 Jan 1997 | 31 Dec 2015 | See Hirpa et al. (2018) | See ECMWF (2018)[1] Balsamo et al. (2009)[2] van der Knijff et al. (2010)[3] |
| Reforecasts H-TESSEL Lisflood_c | ECMWF 20-year control Reforecasts | H-TESSEL | ~$0.28^0$ (~31 km) | Lisflood | $0.10^0$ (~10 km) | Daily | 01 Jan 1997 | 31 Dec 2015 | See Hirpa et al. (2018) | See ECMWF (2017)[1] Balsamo et al. (2009)[2] van der Knijff et al. (2010)[3] |
| ERA-I Land H-TESSEL CaMa-Flood | ERA-I Land | H-TESSEL | ~$0.75^0$ (~80 km) | CaMa-Flood | $0.25^0$ (~25 km) | Daily | 01 Jan 1997 | 31 Dec 2015 | None | Balsamo et al. (2015)[1] Balsamo et al. (2009)[2] Yamazaki et al. (2011)[3] |
| ERA-I Land PCR-GLOBWB CaMa-Flood | ERA-I Land | PCR-GLOBWB | ~$0.50^0$ (~50 km) | CaMa-Flood | $0.25^0$ (~25 km) | Daily | 01 Jan 1997 | 31 Dec 2015 | None | Balsamo et al. (2015)[1] Sutanudjaja et al. (2018)[2] Yamazaki et al. (2011)[3] |
| TRMM CRESTEF5 | TMPA 3B42 v7. Real-time | EF5/CREST | ~$0.25^0$ (~25 km) | EF5/CREST | $0.05^0$ (~5 km) | Daily | 01 Jan 2003 | 31 Dec 2015 | None | Huffman et al. (2007)[1] Wang et al. (2011)[2] Clark et al. (2016)[3] |

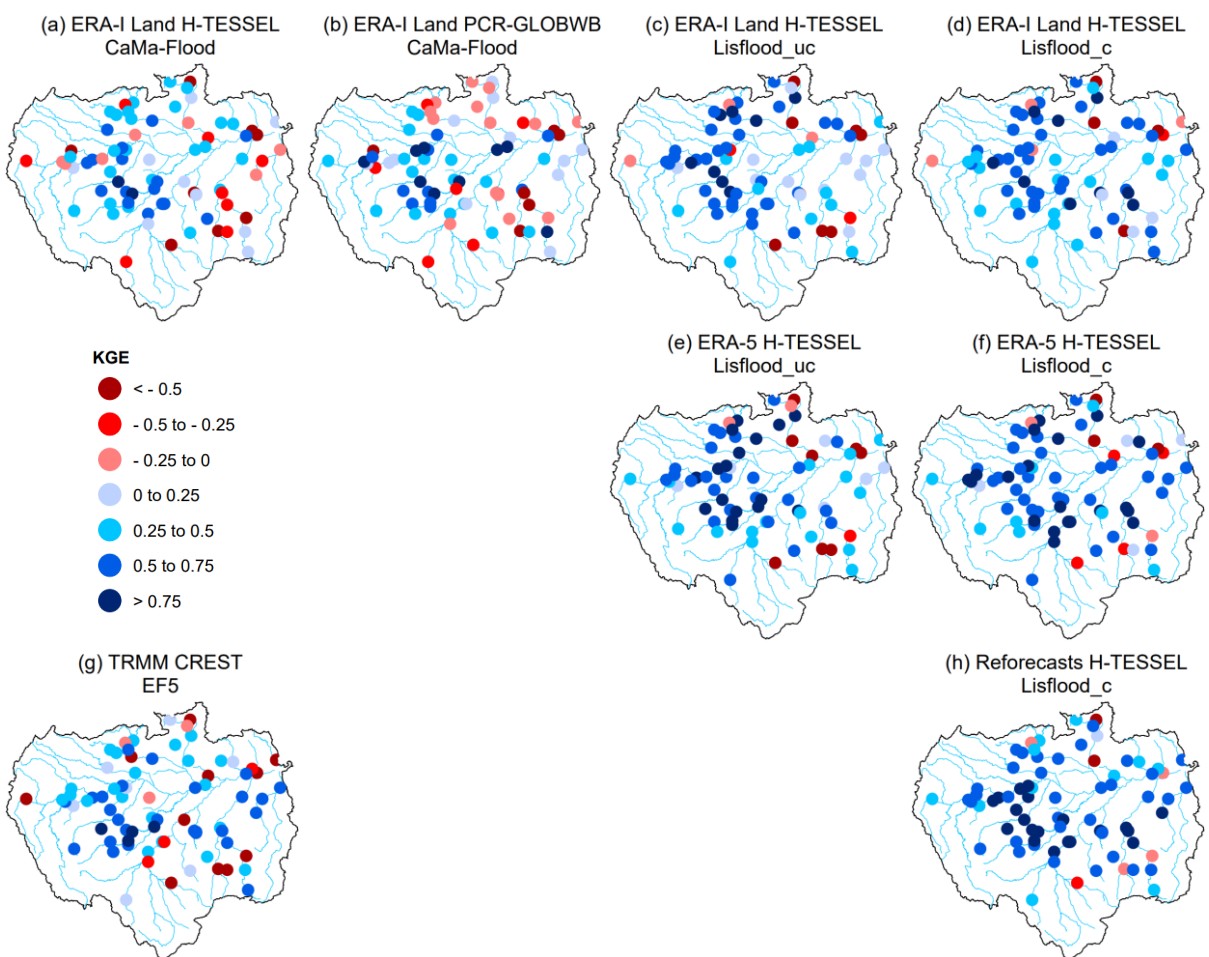

**Figure 2: Full Kling Gupta Efficiency (KGE) scores at the 75 hydrological gauging stations for all simulations. For the period 1997-2015 and 2004-2015 for the Coupled Routing and Excess Storage, Ensemble Framework for Flash Flood Forecasting (CREST EF5) run (g). Values greater than 0.75 are considered to indicate good performance (i.e. dark blue circles). To allow for easier model comparisons, plots are arranged by the different precipitation datasets (rows) and routing models (columns) with the exception of CREST EF5 (g). For example, the final column consists of model runs using the calibrated Lisflood routing model.**

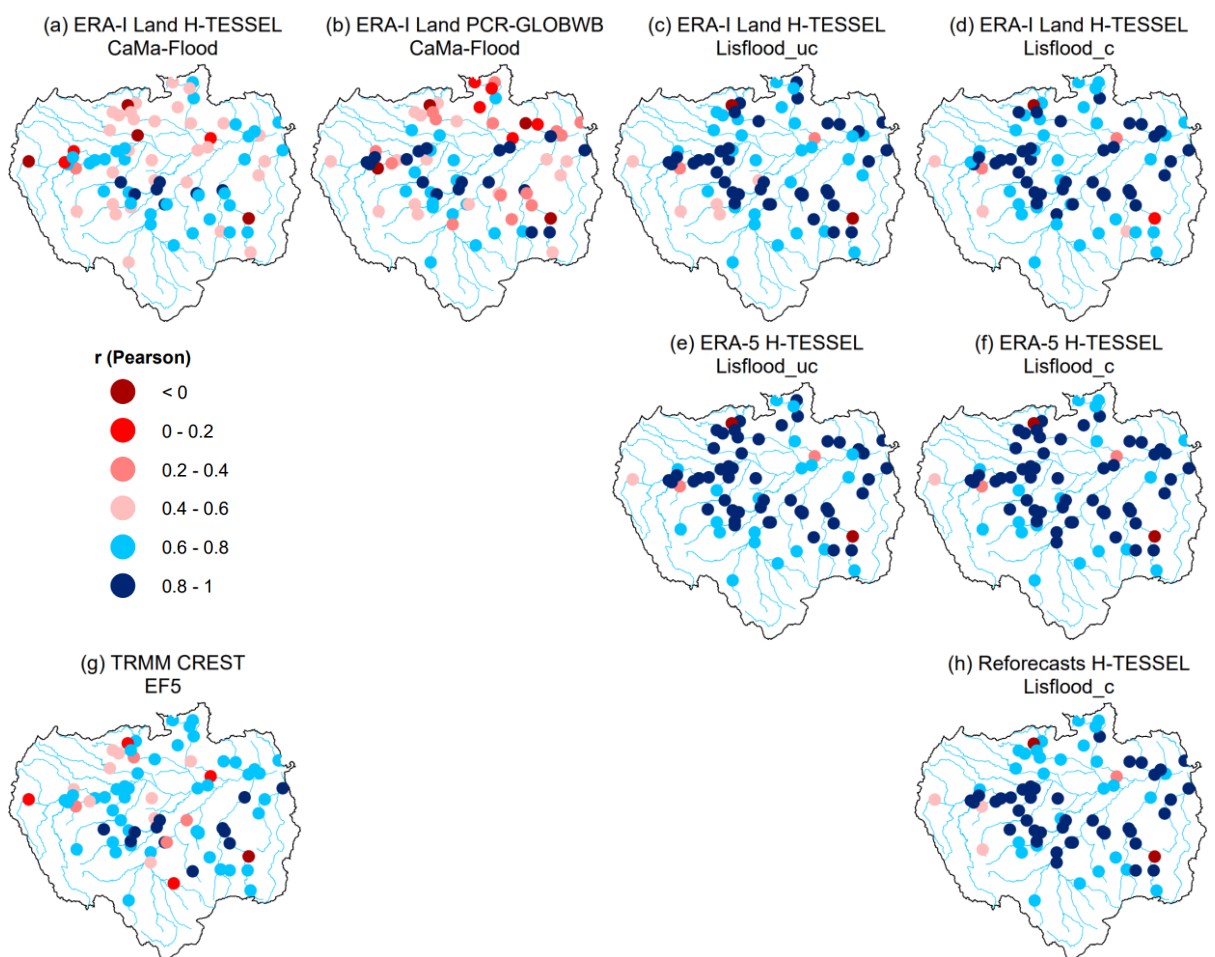

**Figure 3: Correlation component (Pearson's) of the Kling Gupta Efficiency (KGE) at the 75 hydrological gauging stations for all simulations. For the period 1997-2015 and 2004-2015 for the Coupled Routing and Excess Storage, Ensemble Framework for Flash Flood Forecasting (CREST EF5) run (g). Values greater than 0.6 are considered skilful (i.e. blue circles).**

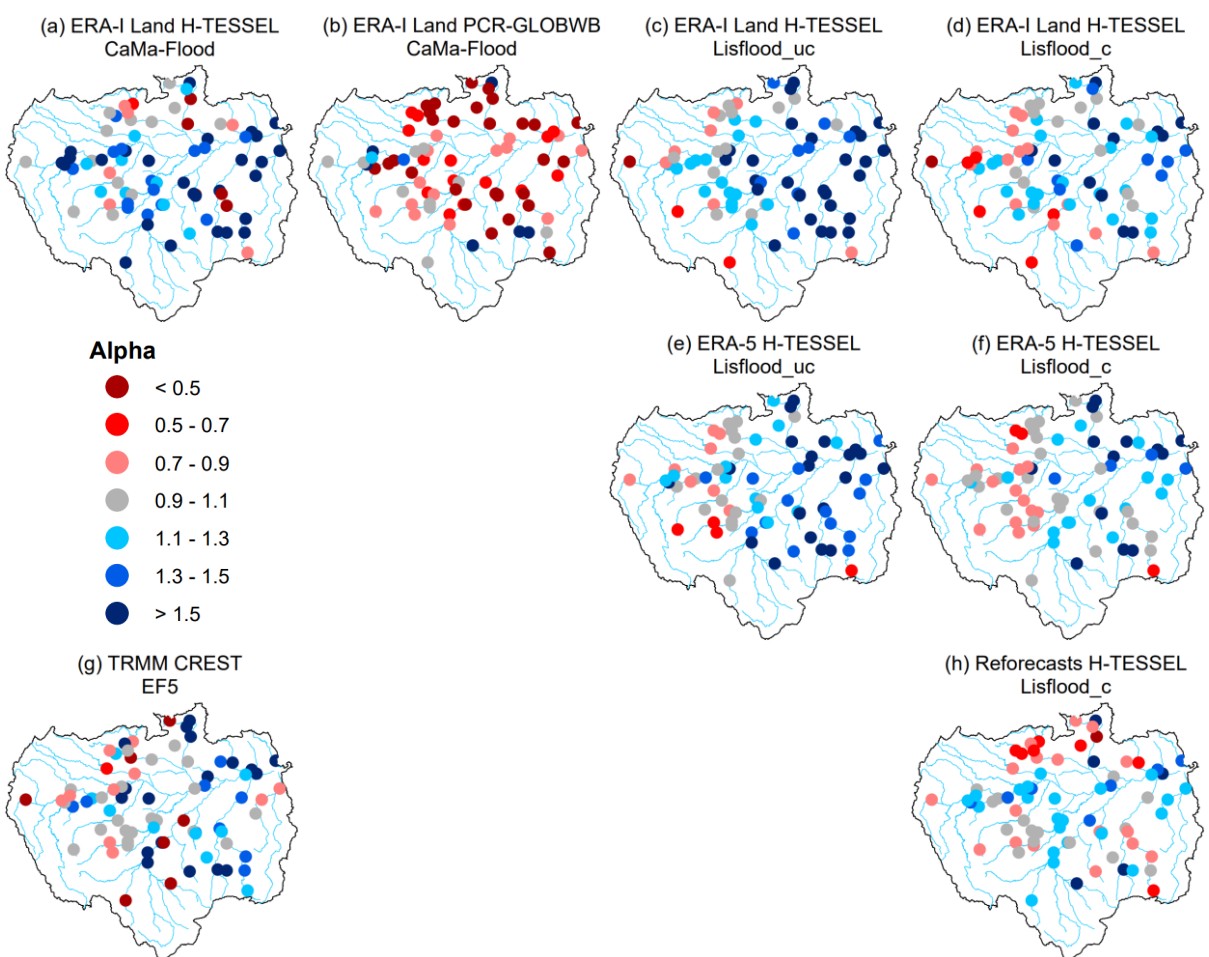

**Figure 4: Alpha (i.e. variability ratio) component of the Kling Gupta Efficiency (KGE) at the 75 hydrological gauging stations for all simulations. For the period 1997-2015 and 2004-2015 for the Coupled Routing and Excess Storage, Ensemble Framework for Flash Flood Forecasting (CREST EF5) run (g). Blue circles indicate that the variability in the simulated time series is higher than that of the observed, while red circles show the opposite effect. Values closer to one indicate better model performance (i.e. grey circles).**

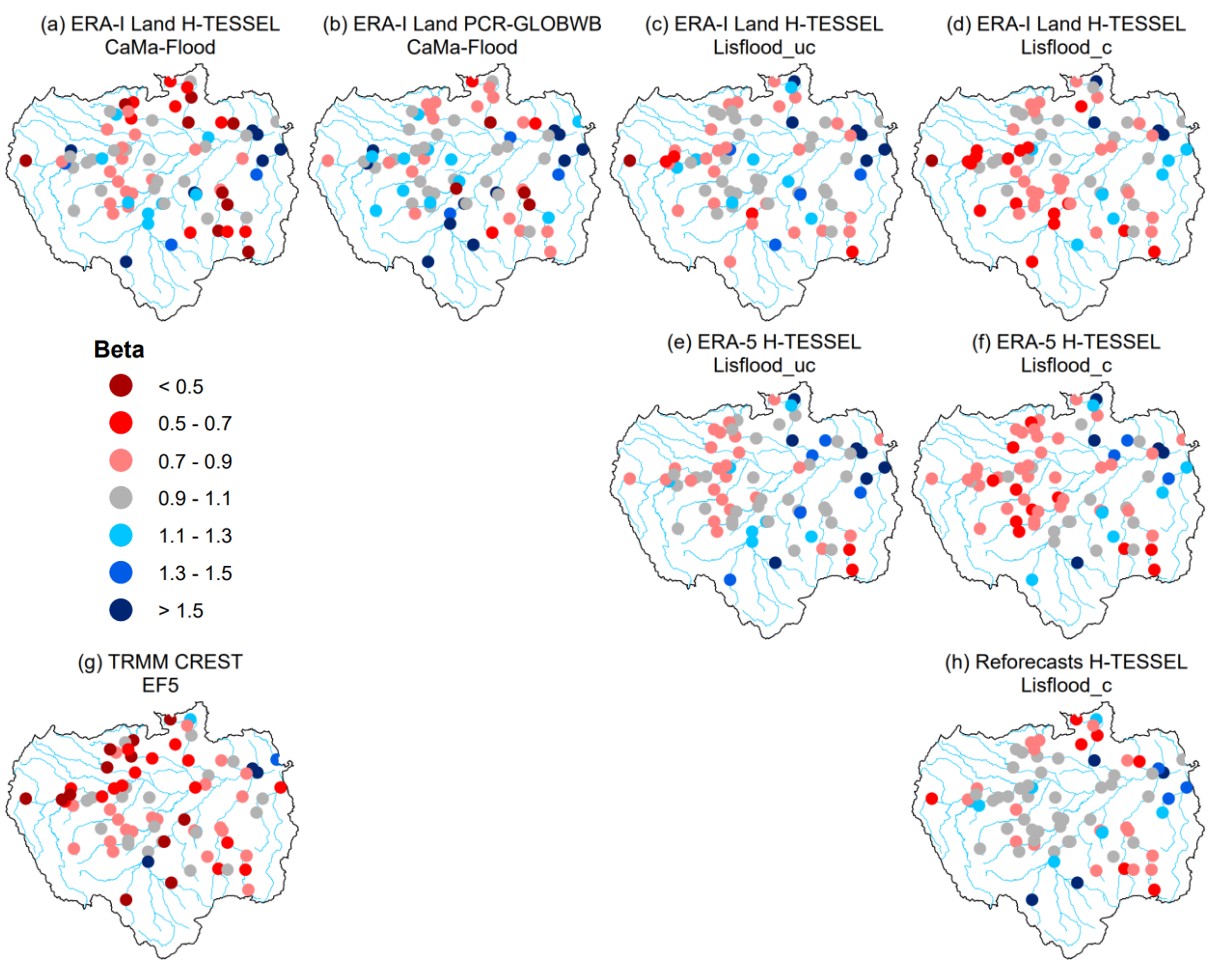

**Figure 5: Beta (i.e. bias ratio) component of the Kling Gupta Efficiency (KGE) at the 75 hydrological gauging stations for all simulations. For the period 1997-2015 and 2004-2015 for the Coupled Routing and Excess Storage, Ensemble Framework for Flash Flood Forecasting (CREST EF5) run (g). Blue circles indicate that the bias in the simulated time series is higher than that of the observed, while red circles show the opposite effect. Values closer to one indicate better model performance (i.e. grey circles).**

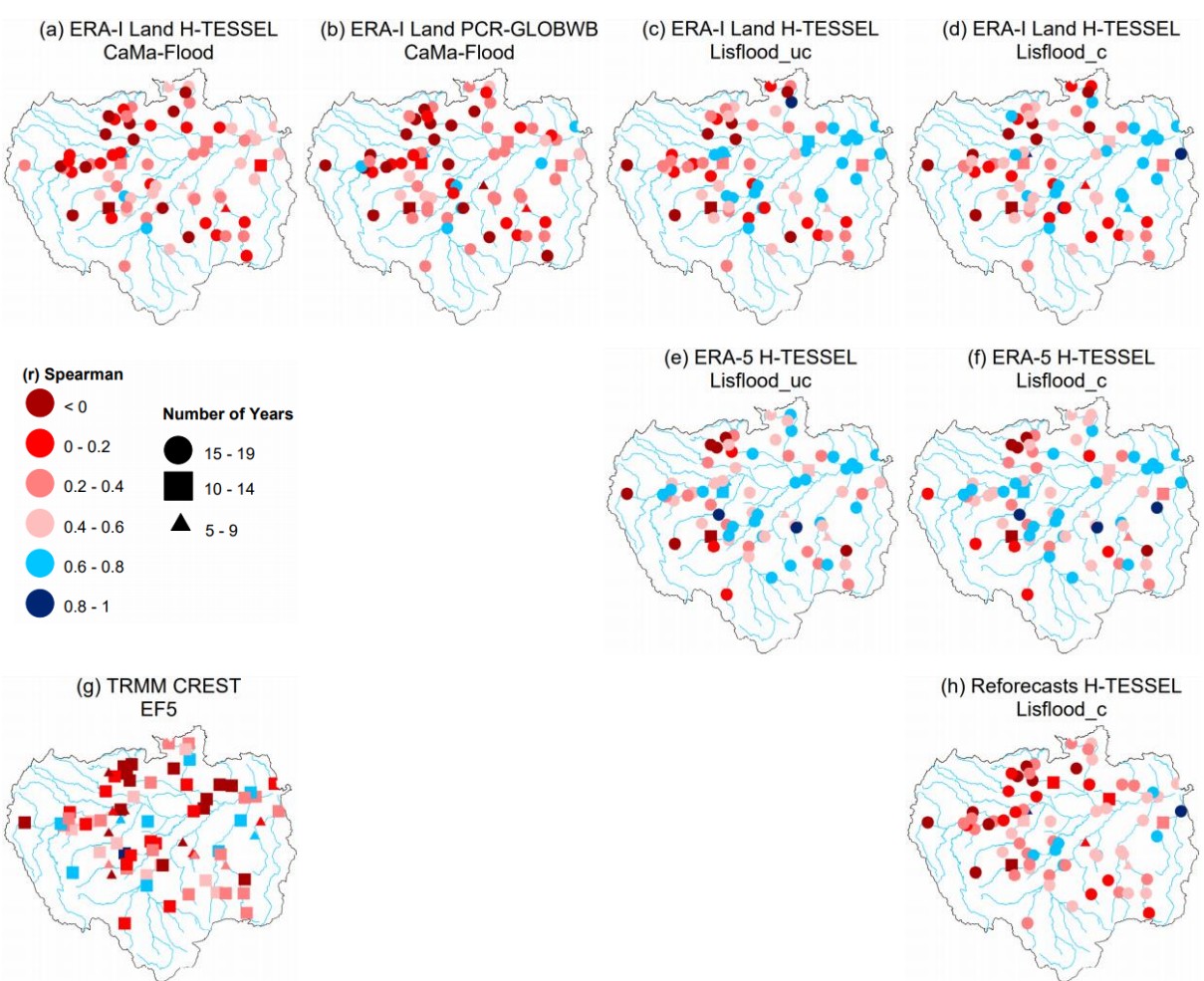

**Figure 6: Spearman's ranked correlation coefficients for observed against simulated annual maximum discharge values at the 75 hydrological gauging stations for all simulations. For the period 1997-2015 and 2004-2015 for the Coupled Routing and Excess Storage, Ensemble Framework for Flash Flood Forecasting (CREST EF5) run (g). Values exceeding 0.6 are considered skilful (i.e. blue shapes). Number of overlapping years of data between observations and simulations are denoted by different shapes. A triangle represents 5-9 years, a square 10-14 years and a circle 15-19 years of overlapping data.**

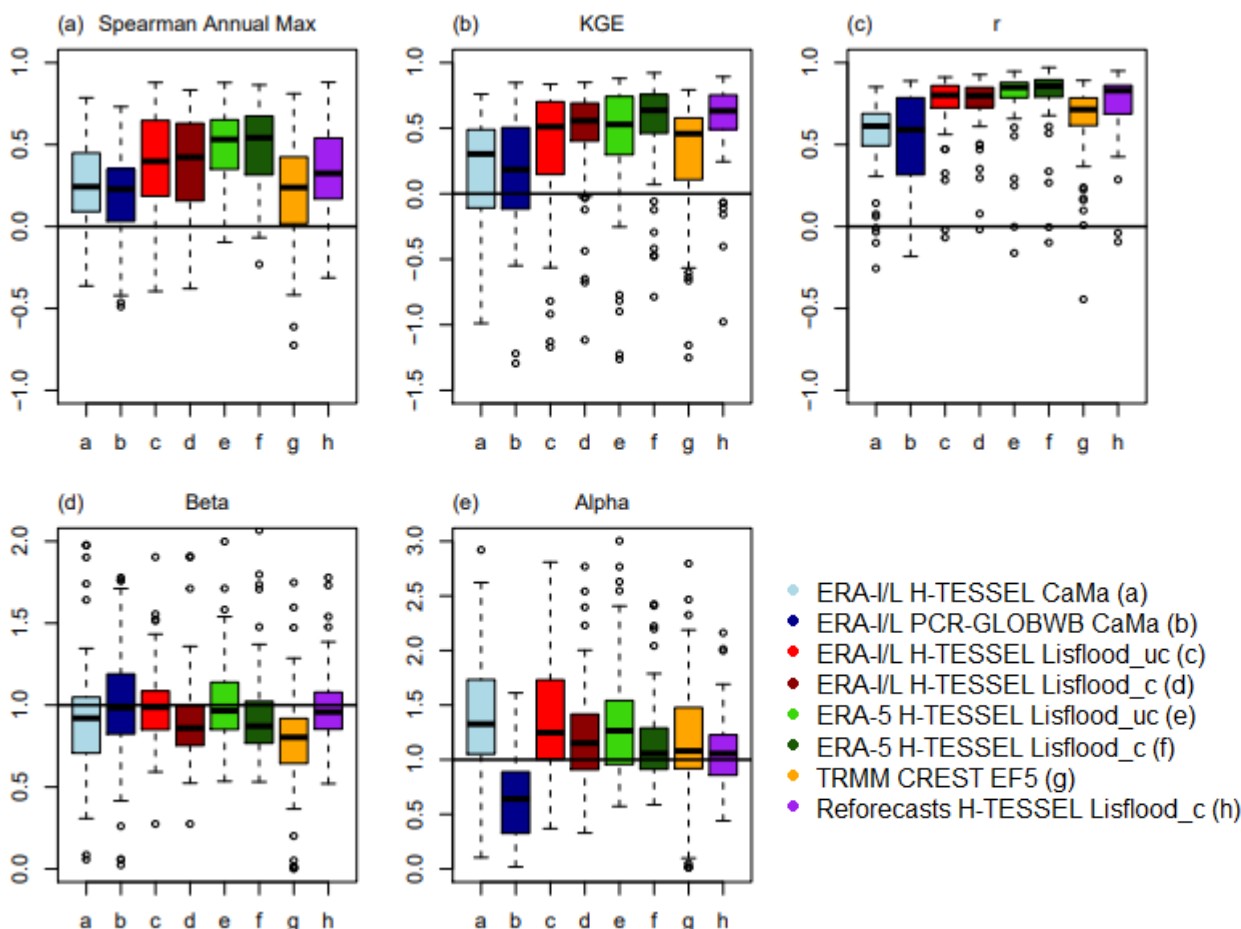

**Figure 7: Boxplots showing the distribution of scores for the (a) Spearman annual maximum correlation (b) Kling Gupta Efficiency (KGE), (c) KGE Pearson's coefficient, (d) KGE beta and (e) KGE alpha, for all simulations. For the period 1997-2015.**

**Table 2: Median scores for the 75 hydrological gauging stations for all metrics.**

| Model Runs | Spearman Annual Max Correlations | KGE | r (Pearson's) | Beta | Alpha |
|---|---|---|---|---|---|
| ERA-Interim Land H-TESSEL CaMa-Flood | 0.24 | 0.30 | 0.61 | 0.92 | 1.33 |
| ERA-Interim Land PCR-GLOBWB CaMa-Flood | 0.23 | 0.18 | 0.59 | 0.98 | 0.64 |
| ERA-Interim Land H-TESSEL Lisflood_uc | 0.40 | 0.51 | 0.80 | 0.99 | 1.25 |
| ERA-Interim Land H-TESSEL Lisflood_c | 0.42 | 0.56 | 0.80 | 0.86 | 1.15 |
| ERA-5 H-TESSEL Lisflood_uc | 0.53 | 0.63 | 0.85 | 0.97 | 1.26 |
| ERA-5 H-TESSEL Lisflood_c | 0.54 | 0.64 | 0.86 | 0.87 | 1.06 |
| TRMM CREST EF5 | 0.24 | 0.46 | 0.71 | 0.80 | 1.08 |
| Reforecasts H-TESSEL Lisflood_c | 0.32 | 0.63 | 0.83 | 0.96 | 1.06 |
| Median across models | 0.35 | 0.50 | 0.78 | 0.91 | 1.11 |

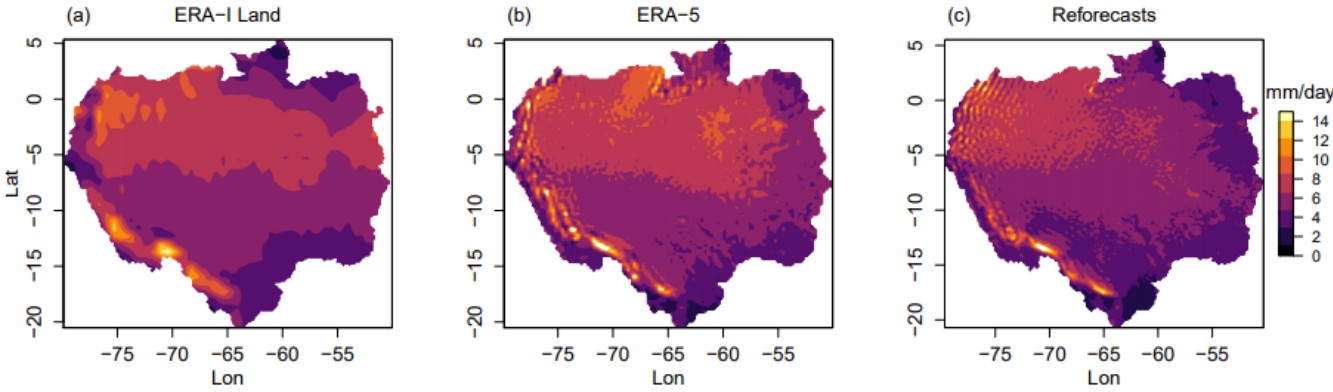

**Figure 8: Mean daily precipitation totals throughout the Amazon basin. For (a) ERA-Interim Land, (b) ERA-5 and (c) the European Centre for Medium-range Weather Forecasts (ECMWF) 20-year reforecasts. For the period 1997-2015.**

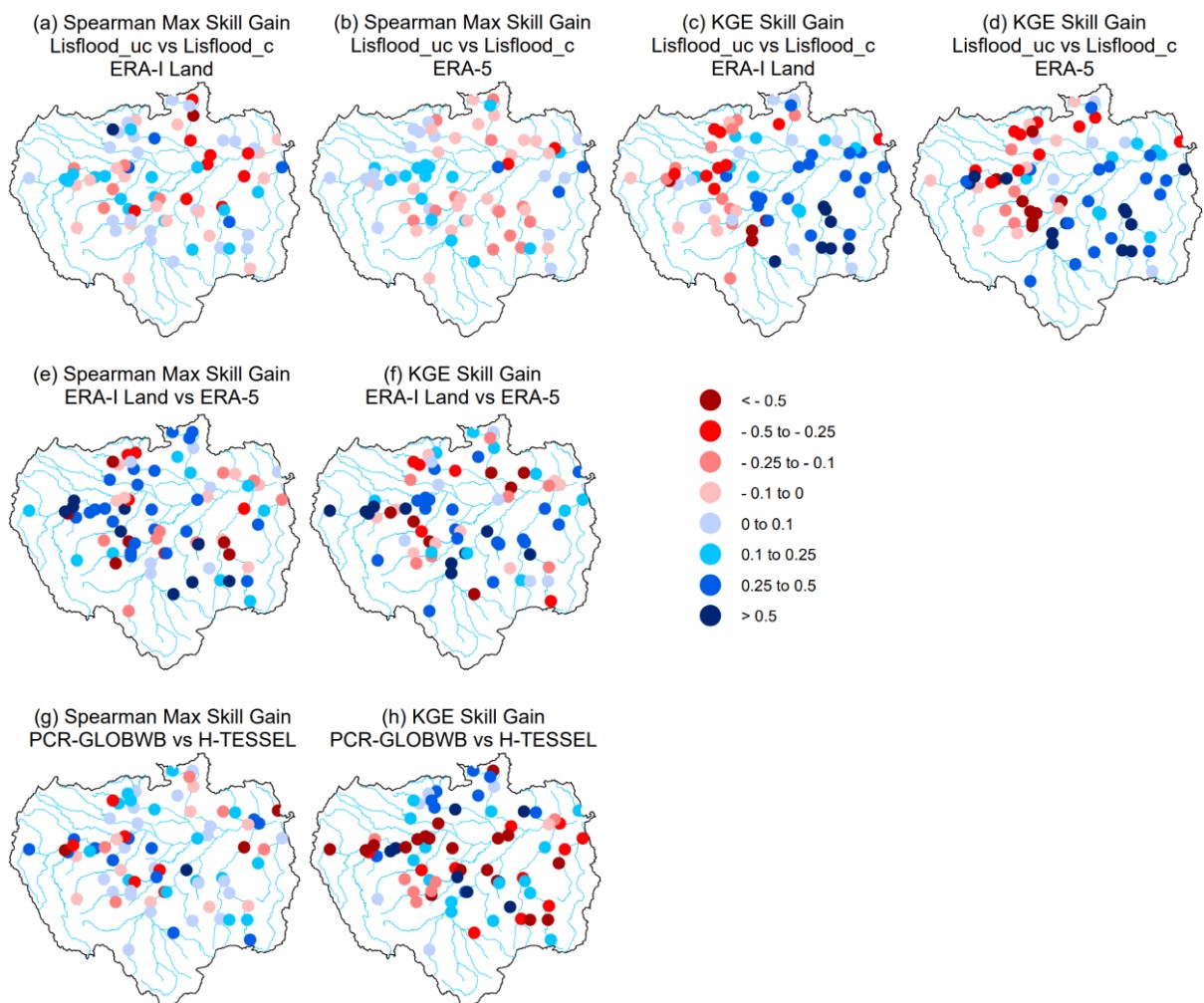

**Figure 9: Relative improvement in skill at each gauging station for Spearman annual maximum correlations and Kling Gupta Efficiency (KGE) values (i.e. skill scores). (a-d) show relative gain or loss in skill when using the calibrated Lisflood run (Lisflood_c) relative to the uncalibrated model run (Lisflood_uc), using precipitation forcing from both ERA-Interim Land and ERA-5. (e-f) shows the relative gain or loss in skill when using ERA-5 as opposed to ERA-Interim Land. (g-h) shows the relative gain or loss in skill when using the Land Surface Model (LSM), the Hydrology-Tiled ECMWF Scheme for Surface Exchanges over Land (H-TESSEL) compared to the hydrological model, PCRaster Global Water Balance (PCR-GLOBWB). All scores are calculated using the skill scores in Eq. (1). Red circles indicate a decrease in skill, whereas blue circles represent an increase.**

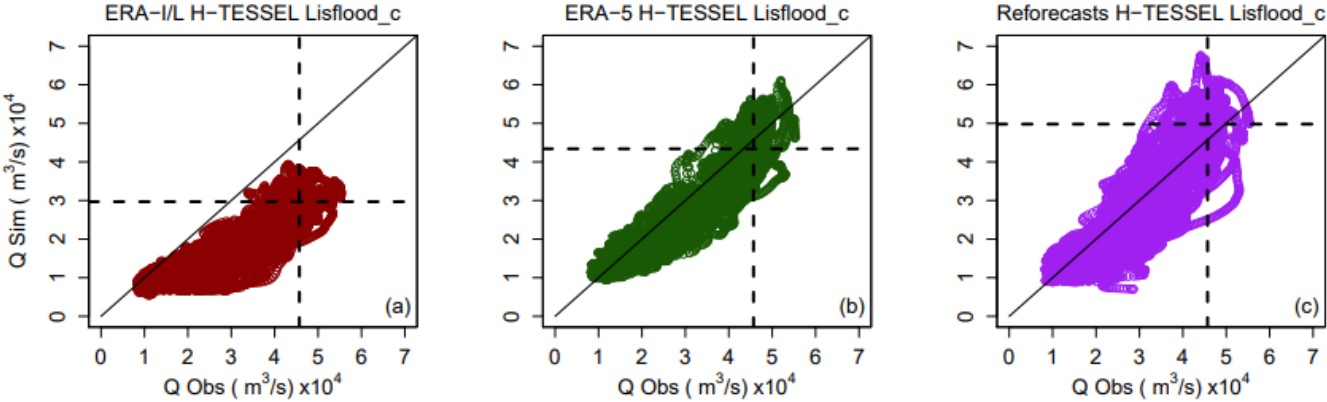

**Figure 10: Scatterplots of observed against simulated river flow at the Tamshiyacu gauging site, Peru (*4*). For (a) ERA-Interim Land, (b) ERA-5 and (c) the European Centre for Medium-range Weather Forecasts (ECMWF) 20-year reforecasts forced through the calibrated Lisflood routing model. Dashed black lines indicate the observed and simulated 90[th] percentile of river flow. For the period 1997-2015.**