# Peer review of "Assessing the performance of global hydrological models for capturing peak river flows in the Amazon Basin"

_Hydrology and Earth System Sciences, 2019_

## Referee Comment (RC1) · Andrew Newman (Referee) · 1 May 2019

General comments:

This paper examines eight global hydrologic models (GHMs) across the Amazon basin for several basic performance metrics. The model configurations allow for somewhat limited comparisons of coarse granularity methodological choices (e.g. routing model, input forcing data) across the performance metrics for a subset of the GHMs. The submission is easy to read and logically organized with clearly stated objectives. The overall conclusion that the precipitation dataset may be the most important for essentially uncalibrated GHMs and that calibration of the routing model using daily KGE as

the objective function has limited effectiveness, particularly for floods, intuitively make sense and agree with the presented results. While not particularly novel, they are useful conclusions to reiterate to the GHM model development and user community.

Specific comments:

1) This paper suffers from the same issues all macroscale intercomparison studies do. In general the analysis is somewhat superficial and the conclusions regarding which modeling component is best (e.g. routing or hydrologic model) are limited to that very coarse scale. Also, while I do appreciate that the authors describe their model selection efforts, it remains that only a subset of the model decision matrix is populated in an ad-hoc manner. This limits comparisons and conclusions to specific subsets of models depending on the question asked.

It may be beneficial for the authors to add some statements describing the limitations of these types of studies. For example, question 3.3 examines routing models, but the comparison is for one hydrologic model and one precipitation dataset only. The conclusions in this section are reasonable for that specific comparison; the results may change given another modeling chain upstream of the routing model.

When discussing future work/paths forward, it may be helpful to talk about increasing the granularity at which modeling decisions are tested and filling out the model decision matrix in a more quantitative way so that more generalized conclusions can be made.

2) There are references relevant to the routing model calibration discussion the authors should consider. Their conclusion that performance is improved for metrics more closely related to the objective function is correct, however the statements on page 17, lines 13-15, and again on page 18, lines 4-5 need further elaboration.

Gupta et al. (2009) and Mizukami et al. (2019) discuss in detail how squared error metrics relate to high flow performance. Mizukami et al. (2019) also show how an application specific metric, annual peak flow bias (APFB) can improve model performance for that specific metric, but at the expense of decreased performance in related metrics. This is directly relevant to the final sentence of the conclusions section, and it would be good to note that some application specific metrics could degrade model performance for other parts of the hydrograph, so thoughtful consideration to the full use of the modeling system should be given when performing parameter optimization.

Sincerely, Andrew Newman

References:

Gupta, H. V., Kling, H., Yilmaz, K. K., and Martinez, G. F.: Decomposition of the mean squared error and NSE performance criteria: Implications for improving hydrological modelling, J. Hydrol., 377, 80–91, https://doi.org/http://dx.doi.org/10.1016/j.jhydrol.2009.08.003, http://www.sciencedirect.com/science/article/pii/S0022169409004843, 2009.

Mizukami, N., Rakovec, O., Newman, A., Clark, M., Wood, A., Gupta, H., and Kumar, R.: On the choice of calibration metrics for high flow estimation using hydrologic models, Hydrol. Earth Syst. Sci. Discuss., https://doi.org/10.5194/hess-2018-391, in review, 2018.

---

## Referee Comment (RC2) · Gemma Coxon (Referee) · 1 May 2019

This manuscript assesses the performance of eight different global hydrological models for 75 gauging stations across the Amazon basin. It assesses the ability of the model to reproduce daily and peak river flows and finds interesting regional differences in model performance. It concludes that the rainfall product is the dominant control on model performance.

Overall, I enjoyed reading the paper. It is well-written and logically structured, with excellently formatted plots. While the analysis and results (as Reviewer 1 also pointed out) are not ground-breaking, the work represents a potentially useful contribution to

the application of GHMs and their use for flood risk assessment and forecasting. For the results to have wider applicability, I think the authors should consider adding some additional analyses to strengthen their conclusions. Many reasons for good/bad model performance are postulated in the results section but the analysis isn't in-depth enough to draw strong conclusions (other than rainfall is a dominant control on model performance). The wider applicability of the results also needs to be better highlighted in the conclusions.

My main comments are detailed below.

Comments

1. The authors conclude that rainfall is (unsurprisingly) the dominant control on model performance. If the choice of precipitation dataset is the most influential on model performance then I would have expected a little more analysis on the precipitation datasets themselves. I appreciate that more involved analysis looking at the seasonal characteristics of the rainfall patterns is probably beyond this paper, however a figure of mean annual rainfall across all the catchments for each rainfall product would be useful for the reader to better understand how and why model performance may vary between the different models across the Amazon basin.

2. P10 L23 "An average of 81% of stations are considered skillful (i.e. > 0) for the KGE metric". I would not consider a KGE score just greater than 0 as 'skillful'. You should be more specific here about what a KGE score greater than zero represents if you are using it as a benchmark to define whether a model is skilful or not. Unlike NSE, a KGE score of zero does not represent the mean streamflow response.

3. P10 L27 (and elsewhere). Your 'average' station scores could be skewed here by particularly low values of KGE – it may be better to report the median station scores here instead.

4. Section 3.1. I liked your analysis of the relationship between model performance

and dams and Figure 1b. However, as you note, this doesn't fully explain regional differences in model performance. Are there any other catchment characteristics that may also explain good/poor model performance? I would calculate and add some additional catchment characteristics such as mean rainfall, mean pet, elevation, geology (as you also focus on groundwater parameterisation) to Figure 1 to better analyse these regional differences in model performance and strengthen the conclusions of the paper.

5. Section 3.1. In addition to comment 4 – do any of the GHMs include schemes to represent dams/reservoirs? Do you see any differences in model performance for models that do include these human influences over models that do not?

6. Conclusions L28-29. You state that "The implications of these results suggest that the choice of precipitation dataset is the most influential component of the GHM set-up in terms of our ability to recreate annual maximum river flows in the Amazon basin". Can you say anything more about what type or spatial resolution of precipitation dataset should be used to better reproduce annual maximum flows? This would help to guide future studies on modelling peak flows in the Amazon basin.

---

## Author Comment (AC1) · 9 May 2019

Referee comments are highlighted in bold with our response in normal font

**General comments:**

**This paper examines eight global hydrologic models (GHMs) across the Amazon basin for several basic performance metrics. The model configurations allow for somewhat limited comparisons of coarse granularity methodological choices (e.g. routing model, input forcing data) across the performance metrics for a subset of the GHMs. The submission is easy to read and logically organized with clearly stated objectives. The overall conclusion that the precipitation dataset may be the most important for essentially uncalibrated GHMs and that calibration of the routing model using daily KGE as the objective function has limited effectiveness, particularly for floods, intuitively make sense and agree with the presented results. While not particularly novel, they are useful conclusions to reiterate to the GHM model development and user community.**

We thank Dr. Newman for his useful evaluation of the manuscript and suggestions for improvement. We will address these in the revised manuscript, as according to our responses to each comment below.

**Specific comments:**

**1) This paper suffers from the same issues all macroscale intercomparison studies do. In general the analysis is somewhat superficial and the conclusions regarding which modeling component is best (e.g. routing or hydrologic model) are limited to that very coarse scale. Also, while I do appreciate that the authors describe their model selection efforts, it remains that only a subset of the model decision matrix is populated in an adhoc manner. This limits comparisons and conclusions to specific subsets of models depending on the question asked.**

**It may be beneficial for the authors to add some statements describing the limitations of these types of studies. For example, question 3.3 examines routing models, but the comparison is for one hydrologic model and one precipitation dataset only. The conclusions in this section are reasonable for that specific comparison; the results may change given another modeling chain upstream of the routing model. When discussing future work/paths forward, it may be helpful to talk about increasing the granularity at which modeling decisions are tested and filling out the model decision matrix in a more quantitative way so that more generalized conclusions can be made.**

Overall, we agree with the limitation of comparing only macroscale features between each GHM (e.g. routing model/precipitation forcing) and the suggestion to explicitly highlight some of these limitations. As Dr. Newman mentions, we do highlight the reasoning behind the model and comparison selections in the manuscript with acknowledgment to alternative methodologies.

We plan to change the sub-heading for Sect. 3.6 to "Limitations and future work" by discussing the limitations of using a macroscale intercomparison approach. We will add to the discussion the possibility to expand the study by increasing the granularity at which modelling decisions are tested; thereby allowing more generalised conclusions to be made.

**2) There are references relevant to the routing model calibration discussion the authors should consider. Their conclusion that performance is improved for metrics more closely related to the objective function is correct, however the statements on page 17, lines 13-15, and again on page 18, lines 4-5 need further elaboration. Gupta et al. (2009) and Mizukami et al. (2019) discuss in detail how squared error metrics relate to high flow performance. Mizukami et al. (2019) also show how an application specific metric, annual peak flow bias (APFB) can improve model**

**performance for that specific metric, but at the expense of decreased performance in related metrics. This is directly relevant to the final sentence of the conclusions section, and it would be good to note that some application specific metrics could degrade model performance for other parts of the hydrograph, so thoughtful consideration to the full use of the modeling system should be given when performing parameter optimization.**

We thank Dr. Newman for this very useful suggestion. We agree that the original conclusion would benefit from further elaboration and evidence from previous works.

We will incorporate both the suggested references to support our discussions and conclusions. Specifically:

i) By commenting on the suitability of square error related metrics for model calibration when the application requires robust performance for high flows (in Sect. 3.6)

ii) By providing the example of increased model performance when using application specific metrics (i.e. Annual Peak Flow Bias), as in Mizukami et al. (in review)

iii) By discussing the need to carefully consider the metric used in calibration, with the possibility of a loss in skill for other related metrics upon evaluation when using application specific metrics (Sect 3.6 and Sect. 4)

---

## Author Comment (AC2) · 9 May 2019

Referee comments are highlighted in bold with our response in normal font

**Overall comment:**

**This manuscript assesses the performance of eight different global hydrological models for 75 gauging stations across the Amazon basin. It assesses the ability of the model to reproduce daily and peak river flows and finds interesting regional differences in model performance. It concludes that the rainfall product is the dominant control on model performance. Overall, I enjoyed reading the paper. It is well-written and logically structured, with excellently formatted plots. While the analysis and results (as Reviewer 1 also pointed out) are not ground-breaking, the work represents a potentially useful contribution to the application of GHMs and their use for flood risk assessment and forecasting. For the results to have wider applicability, I think the authors should consider adding some additional analyses to strengthen their conclusions. Many reasons for good/bad model performance are postulated in the results section but the analysis isn't in-depth enough to draw strong conclusions (other than rainfall is a dominant control on model performance). The wider applicability of the results also needs to be better highlighted in the conclusions.**

We thank Dr. Coxon for her useful evaluation of the manuscript and kind comments. The specific comments are particularly helpful and will improve the overall quality of the paper. We will address these in the revised manuscript, as according to our responses to each comment below.

**Specific comments:**

**1) The authors conclude that rainfall is (unsurprisingly) the dominant control on model performance. If the choice of precipitation dataset is the most influential on model performance then I would have expected a little more analysis on the precipitation datasets themselves. I appreciate that more involved analysis looking at the seasonal characteristics of the rainfall patterns is probably beyond this paper, however a figure of mean annual rainfall across all the catchments for each rainfall product would be useful for the reader to better understand how and why model performance may vary between the different models across the Amazon basin.**

We agree with Dr. Coxon that further evaluation of the rainfall products would be beneficial in further understanding differences in model performance given the importance of rainfall in the ability to accurately represent peak flows.

We propose to incorporate a Figure showing mean annual rainfall across the basin for each rainfall product in Sect. 3.4.

**2) P10 L23 "An average of 81% of stations are considered skillful (i.e. > 0) for the KGE metric". I would not consider a KGE score just greater than 0 as 'skillful'. You should be more specific here about what a KGE score greater than zero represents if you are using it as a benchmark to define whether a model is skilful or not. Unlike NSE, a KGE score of zero does not represent the mean streamflow response.**

We thank Dr. Coxon for this useful comment. We propose to remove such statements from the manuscript. A similar study (Thiemig et al., 2018) provides a breakdown of KGE (though for the modified version of KGE) results into "good" (KGE > 0.75), "intermediate" (0.75 > KGE > 0.5, "poor"

(0.5 > KGE > 0) and "very poor (KGE < 0)" categories. We now plan to use these categories as the benchmark but generally refer to KGE performance as being better as values approach one.

Thiemig, V., Bisselink, B., Pappenberger, F., & Thielen, J.: A pan-African medium-range ensemble flood forecast system. Hydrol. Earth Syst. Sci, 19, 3365-3385, https://doi.org/10.5194/hess-19-1-2015, 2015.

**3) P10 L27 (and elsewhere). Your 'average' station scores could be skewed here by particularly low values of KGE – it may be better to report the median station scores here instead.**

We again thank Dr. Coxon for raising this relevant point. We will change all reported mean values to median values.

**4) Section 3.1. I liked your analysis of the relationship between model performance and dams and Figure 1b. However, as you note, this doesn't fully explain regional differences in model performance. Are there any other catchment characteristics that may also explain good/poor model performance? I would calculate and add some additional catchment characteristics such as mean rainfall, mean pet, elevation, geology (as you also focus on groundwater parameterisation) to Figure 1 to better analyse these regional differences in model performance and strengthen the conclusions of the paper.**

As Dr. Coxon mentions, damming in the Amazon is one of many catchment characteristics that could influence model performance. We agree that by including additional characteristics such as rainfall and geology, stronger conclusions can be drawn on regional model performance. For example, Paiva et al. (2013) indicate that poorer model performance in western tributaries could be associated with the poor representation of mountainous rainfall when using satellite-based products such as TRMM. While in Sect 3.1 we hypothesis that the underestimation of flows in stations located in the south eastern Amazon could be due to a poor representation of the South Atlantic Convergence Zone (SACZ). By adding rainfall maps to Fig. 1 or elsewhere (e.g. Sect. 3.4) these statements will be strengthened.

de Paiva, R. C. D., Buarque, D. C., Collischonn, W., Bonnet, M. P., Frappart, F., Calmant, S., & Mendes, C. A. B.: Large-scale hydrologic and hydrodynamic modeling of the Amazon River basin, Water Resour. Res., *49*, 1226-1243, https://doi.org/10.1002/wrcr.20067,2013, 2013.

**5) Section 3.1. In addition to comment 4 – do any of the GHMs include schemes to represent dams/reservoirs? Do you see any differences in model performance for models that do include these human influences over models that do not?**

This is an interesting point. Model runs which incorporate the Lisflood routing model (i.e. those similar to the current GloFAS set-up) represent a total of 464 lakes and 667 reservoirs obtained from global databases. Very few reservoirs however are represented in the Amazon basin for Lisflood with the majority of lakes positioned along the main Amazon River. The locations and importance of these to the modelling system has been analysed by Zajac et al. (2017) for the entire globe with the effects of reservoirs on extreme high flows deemed substantial and widespread in the global domain. All other simulations do not account for dams/reservoirs and thus comparison between these runs could strengthen our understanding of why some models perform particularly well or poor at the few stations affected.

We will add information regarding the representation of lakes and reservoirs within Lisflood in Sect. 2.2.3. In addition, we will investigate the potential benefits or losses in skill by comparing model performance at stations located near to any reservoirs between Lisflood and non-Lisflood simulations. Although the number of stations effected will be limited it could still provide useful information.

Zajac, Z., Revilla-Romero, B., Salamon, P., Burek, P., Hirpa, F. A., & Beck, H.: The impact of lake and reservoir parameterization on global streamflow simulation, J. Hydrol., 548, 552-568, https://doi.org/10.1016/j.jhydrol.2017.03.022, 2017.

**6) Conclusions L28-29. You state that "The implications of these results suggest that the choice of precipitation dataset is the most influential component of the GHM set-up in terms of our ability to recreate annual maximum river flows in the Amazon basin". Can you say anything more about what type or spatial resolution of precipitation dataset should be used to better reproduce annual maximum flows? This would help to guide future studies on modelling peak flows in the Amazon basin.**

We agree that statements regarding the type of spatial resolution required would be beneficial for future studies modelling peak flows in the Amazon basin. However, it would be difficult to make direct statements (e.g. provide a specific spatial resolution) based on our results alone owing to the conclusions being specific to the few directly comparable datasets (i.e. ERA-I Land, ERA-5 and ECMWF 20- year reforecasts forced within the calibrated Lisflood model) used. We can, however, as mentioned in Sect. 3.4, elude to specific cases which may benefit from using certain datasets. For example, those wishing to model peak flows in the Peruvian Amazon may benefit from a dataset where the upper atmosphere is resolved at a higher resolution owing to those particular rivers originating from mountainous regions where rainfall can often be poorly represented (e.g. Paiva et al., 2013).

We plan to expand on our discussions, particularly in Sect. 3.4, to include further findings from studies such as Beck et al. (2017) whereby 22 precipitation products are evaluated across the globe. This could help explain further our results and allows us to strengthen our conclusions on why particular datasets lead to particularly good or bad performance in certain regions.

Beck, H. E., Vergopolan, N., Pan, M., Levizzani, V., van Dijk, A. I., Weedon, G. P., & Wood, E. F.: Global-scale evaluation of 22 precipitation datasets using gauge observations and hydrological modeling. Hydrol. Earth Syst. Sci, 21, 6201-6217, https://doi.org/10.5194/hess-21-6201-2017, 2017.